# Synthetic Aperture Radar Doppler Tomography Reveals Details of Undiscovered High-Resolution Internal Structure of the Great Pyramid of Giza

**Filippo Biondi** [1,†,‡] **and Corrado Malanga** [2,*,‡]

1   Department of Electronic and Electrical Engineering, University of Strathclyde, Glasgow G1 1XW, UK
2   Department of Chemistry and Industrial Chemistry, University of Pisa, Via Giuseppe Moruzzi 13, 56100 Pisa, Italy
*   Correspondence: corrado.malanga@unipi.it; Tel.: +39-342-723-7507
†   Current address: Via Perella 36, 06073 Corciano, Italy.
‡   These authors contributed equally to this work.

**Abstract:** A problem with synthetic aperture radar (SAR) is that due to the poor penetrating action of electromagnetic waves inside solid bodies, the capability to observe inside distributed targets is precluded. Under these conditions, imaging action is provided only on the surface of distributed targets. The present work describes an imaging method based on the analysis of micro-movements on the Khnum-Khufu Pyramid, which are usually generated by background seismic waves. The obtained results prove to be very promising, as high-resolution full 3D tomographic imaging of the pyramid's interior and subsurface was achieved. Khnum-Khufu becomes transparent when observed in the micro-movement domain. Based on this novelty, we have completely reconstructed internal objects, observing and measuring structures that have never been discovered before. The experimental results are estimated by processing series of SAR images from the second-generation Italian COSMO-SkyMed satellite system, demonstrating the effectiveness of the proposed method.

**Keywords:** synthetic aperture radar; doppler frequencies; multi-chromatic analysis; micro motion; pyramid of Khnum-Khufu; sonic images

## 1. Introduction

The Pyramid of Khnum-Khufu, also known as the Great Pyramid of Giza or Cheops, is the oldest and largest of the three main pyramids that are part of the necropolis of Giza (Egypt). The infrastructure is built with blocks of granite, weighing approximately 2.5 t each. Completion of the work is estimated to have taken at least two and a half million blocks, put in place with millimeter precision in a short period of time, estimated at around 15 or 30 years [1]. Despite being one of the oldest and largest monuments on Earth, to date, there is still no common and scientifically established idea on how the pyramids of Egypt were built [2,3]. The Red Sea was the most important Harbor Facilities at the time of King Khufu [4], where an exceptionally well-preserved harbor complex from the Early Old Kingdom at Wadi al-Jarf along the Egyptian coast of the Red Sea has been excavated.

### 1.1. Egyptology Engineering and Ultrasound Introduction

In studying the origin of the pyramids, we believe we should not overlook the existence of ancient mythological writings. A study concerning the myths and folklore of the ancient peoples of the world, highlighting all the similarities between them, was made in [5]. The argument that myths are insignificant—often considered mere stories passed on through generations—has been challenged. The authors are open to the possibility that a technologically more advanced civilization existed before a known timeline, where the existence of various glacial ages [6] prevented the passing down of history. They focus on

the mythical cities mentioned in ancient Indian texts, describing how that subcontinent was an integral part of this [7,8]. However, how the Egyptian Pyramids were built has remained an enduring mystery [9–11]. A theory that the pyramids were cast of cement-like conglomerate made directly in situ using granular limestone aggregates and an alkali–silicate binder is proposed in [12], and evidence is also discussed in [13,14]. In order to obtain an accurate perception of how the pyramids were constructed, various engineering hypotheses were evaluated in [15], noting that in their current form, they lay the foundations for new theories. At present, the general academic consensus is that pyramids served as funerary monuments and burial sites for the pharaohs. However, it is also widely theorized that such infrastructures may have been built for another purpose. On an aseptic panoramic view, many connections can be found between the pyramids, vibrations and many mechanical devices reminiscent of hydraulic systems, resonance chambers and acoustic filters [16,17]. The energy to make the pyramids vibrate can be provided by the natural environment, and the Earth's atmosphere infrasound vibrations may provide the source of such energy [18]. This provides a basis for the discussion of special classes of waves, including mountain Lee-waves, infrasound, progressive waves in the lower atmosphere, and waves in the upper atmosphere and ionosphere [19]. Atmospheric sound models extended to the combination effects of both finite depth ocean and source directivity in both elevation and azimuth angles are studied in [20].

Acoustic waves are widely used in the field of archaeology. The acoustics of three important World Heritage sites such as the five caves in Spain, the Stonehenge stone circle in England and the Paphos Theatre in Cyprus are studied in [21]. Groundwater can influence the geomagnetic field measured in the subsurface. The level of water in the rock determines its electrical conductivity and thus changes the magnitude of the telluric currents induced in the rock by the change in magnetic fields generated in the ionosphere. This can be studied by using several magnetometers at different points in the subsurface. Geomagnetic signals using two magnetometers were successfully monitored in [22] by setting an optimal electrokinetic magnitude signal upper-bound. A methodology used for self-potential and seismic–electromagnetic measurements, both for on-site and laboratory experiments as well as for modeling, is extensively described in [23]. The research also provides the bibliography on studies carried out in hydrology to remotely detect water flows, to deduce their thickness, and to predict their hydraulic conductivity. The observation method discussed also proposes the detection of fractures in wells, which is also useful in trying to study earthquakes. Recent theoretical and experimental studies have produced several unusual and interesting results on the cold fusion of matter experienced on dense lithium [24]. The existence of this exciting propriety of matter relates to zero-point energy estimates that suggest quantum effects play a significant role in shaping the phase diagram of lithium. The vibration-induced property change in the melting and solidifying process of silver nanoparticles with the use of molecular dynamics simulation was found in [25].

The general problem of acoustic wave propagation through parallel paths is addressed by the information theory of two-port telecommunication networks. This allows any mechanical system to be considered as a single element with two gates. Such a circuit is schematically represented with concentrated elements by admittance that have a value compared to the sum of the corresponding instantaneous admittance existing along the parallel paths. In order to calculate all parameters including those representing transmission losses (e.g., the standing wave ratio), in the case of a non-adaptive paths, the theory of transmission lines is well applicable. A Quincke tube acoustic filter therefore uses two parallel paths. In order to optimize the maximum wave propagation through the system, the Quincke tube must be adapted and, through the choice of length and thickness of the ducts, can produce a selective transmission loss, so that it can operate as an acoustic filter. In [26], the transmission loss characteristics of several other variations in duct sizes and lengths are presented together with some very limited experimental data. The phenomenon of pressure pulsations in pipeline systems caused by centrifugal pumps or reciprocating compressors is known to have detrimental effects on industrial applications. An experi-

mental investigation of the attenuation mechanism of a Herschel–Quincke device and its effectiveness in damping pressure pulsations when applied to a resonant piping system had been presented in [27].

The development of mechanical engineering in ancient Egypt through the stone industry was described in [28], covering the period from the Predynastic to the Old Kingdom. The characteristics and innovations of stone vessels available in these periods were analyzed. Conventional sound absorbers can hardly possess the good performance of low-frequency and broadband absorption simultaneously. In order to combine these two functions into one kind of absorbers, the gradually perforated porous materials backed with Helmholtz resonant cavity are proposed in [29]. A strategy to design three-dimensional elastic periodic structures endowed with complete band-gaps, the first of which is ultra-wide, where the top limits of the first two band-gaps are overstepped in terms of wave transmission in the finite structure is proposed in [30]. Thus, subsequent band-gaps are merged, approaching the behavior of a three-dimensional low-pass mechanical filter.

The debate on how the granite blocks could have been transported up the full height of the pyramids is still an open one. To this end, the theory of in situ formation of the blocks by means of a cement mixture has also been formulated. Most synthetic stones can be made from re-agglomerated materials. Starting with a mineral substance such as granite rock or naturally eroded, disintegrated or not-aggregated limestone, it is given a compact structure using a binder, such as a geological glue that agglomerates to bond the mineral particles to each other. The result is a new rock with the same mechanical characteristics as a natural equivalent. Such a technique is supposed to have been used to build the pyramid of Khnum-Khufu [12,31].

### 1.2. Archaeology Heritage Investigation Introduction

Back to the pyramid of Khnum-Khufu, for over a century, it has been known that the beams forming the ceiling of the King's Chamber and those of the first and second Relieving Chambers in the Great Pyramid are cracked. However, the temporal origin of these cracks is still unknown. The results of a 3D virtual reality computer simulation designed to determine precisely when the beams cracked are reported in [32]. Several 3D imaging techniques applied on the Khnum-Khufu pyramid are developed in [33]. Among all the theories formulated to try to explain how the pyramid of Khnum-Khufu was built, there is also the hypothesis of the existence of an internal ramp that goes around the pyramid several times. This theory could prove the fact that the pyramid could have been built in twenty years [34]. Microgravity surveys of the Khnum-Khufu pyramid have been conducted, considering also the general structure of the pyramid. A new interpretation technique for the endoscopy of large finite bodies has been developed in [35]. In order to carry out non-invasive internal scans of the pyramid [36], using electromagnetic waves, it is necessary to use special georadar [37], which however has the limitation of having little penetration inside the granite.

The study of ancient Egyptian monuments attracted the attention of experts all over the world. A recent event that confirms this is the discovery, using muon sensors [38,39], of the presence of a previously unknown cavity located inside the pyramid of Khnum-Khufu. Since this discovery cannot be directly confirmed by drilling, another independent non-destructive method is needed to confirm this discovery and provide an accurate determination of the location and shape of the cavity. A possible holographic radar simulation framework for the detection of openings or other unknown structures of interest is analyzed in [36]. Research [40] for the first time investigated the possibility of using cosmic-ray detectors involved their ability to measure the angle of arrival of penetrating cosmic rays muons with great precision over a large sensitive area. In [41], the authors reported the discovery of a large void (with a cross-section similar to that of the Grand Gallery and a minimum length of 30 m) situated above the Grand Gallery.

The investigation of the microgravimetric measurements on the side of a pyramid could also map the recently discovered "muon chamber" in the Great Pyramid of Khnum-

Khufu in Egypt. In [42], the exploitation of technical capabilities of modern gravimeters is used to perform three-dimensional model calculations with realistic model parameters. A gradiometer survey has been carried out in [43] over a surface area of 100 m × 100 m to achieve the purpose, and the magnetic data were processed using Geoplot software in order to obtain high-quality images of hidden structures inside the Khnum-Khufu pyramid. The results obtained show the presence of interconnected large tomb structures composed of mud-bricks; some other ancient rooms and walls are also present.

A climbing robot called "Djedi" has been designed, constructed, and deployed in [44] to explore shafts of the Queen's chamber within the Great Pyramid. The Djedi robot is based on the concept of inchworm motion and is capable of carrying a long reach drill or snake camera. The robot successfully climbed the southern shaft of the Great Pyramid, deployed its snake camera, and revealed writing not seen for thousands of years. Robot design, including climbing steps in the shaft and lessons learned from experimental deployment, has been designed in [44].

*1.3. Synthetic Aperture Radar Introduction*

Satellite remote sensing is widely used in the field of archaeology [45,46]. Data from the use of SAR to survey the southern Maya plains suggest that large areas were continuously drained by ancient canals that may have been used for intensive cultivation. In agreement with the authors of [47], SAR remote sensing confirmed the existence of the canals. Through excavations and in situ ground surveys, they provided sufficient comparative information. Correlating all the data, it was concluded that the Maya civilization, of the Late Classic period, was firmly based on the intensive and large-scale cultivation of marshy areas. Research [48] found an ideal model configuration, associated with spiral ramps, demonstrating how Egyptians could have built the pyramids. In the past, Synthetic Aperture Radar (SAR) vibrations have been very useful in estimating key vessel characteristics. Research [49,50] proposed a novel strategy to estimate the micro-motion (MM) of ships from SAR images. The proposed approach is for the MM estimation of ships, occupying thousands of pixels, which processes the information generated during the coregistration of several re-synthesized time-domain and not overlapped Doppler sub-apertures of the COSMO-SkyMed satellite single-look complex (SLC) data. The multi-chromatic analysis (MCA), coined for the first time in [51], and performed in the range direction, allows retrieving unambiguous height information on selected pixels, potentially solving the problem of spatial phase unwrapping and absolute phase measurement of interferometric SAR data, under the condition of a sufficient Chirp bandwidth present on both coregistered master and slave SAR SLC images. The authors of [52] propose a new procedure to monitor critical infrastructures such as the Mosul dam, processing COSMO-SkyMed data. The proposed procedure is an in-depth modal assessment based on the MM estimation through a Doppler sub-apertures tracking and the innovation of 90° tilted Doppler-domain MCA. The procedure described above was made available to perform a comprehensive survey of large road bridges, according to [53]. The authors of [52,54,55] successfully formulated a comprehensive procedure to perform structural health monitoring using SAR. The technique allows one to successfully estimate the position and shape of cracks on bridges in order to prevent their collapse. The method is based initially on the persistent scatterers interferometry [56,57] that is also discussed for completeness and validation. The modal analysis has detected the presence of several areas of resonance that could mean the presence of cracks, and the results have shown that the dam is still in a strong destabilization. The article provides an in-depth study of the physical characteristics of vibrations in terms of amplitude, frequency, polarization, and robustness of the estimate in terms of the signal-to-noise ratio of each SAR pixel considered, both through the evaluation of simulated data and by processing real data.

In this paper, we use a new method based on the tomographic reconstruction of MM, with the aim of performing imaging of the principal targets that make the main internal structure of the pyramid visible. We use the similar methods already experimented in [58]

to search for cracks in large infrastructures but not for tomography. The physical principle we use is that of estimating the vibrations captured by the Khnum-Khufu pyramid during the SAR observation time interval. The vibration estimation is completed by evaluating the Doppler centroid anomalies, which is an indispensable parameter that is used during the SAR azimuth focusing process. We use Doppler sub-apertures to estimate the vibrations present on the pyramid. The vibration energy is generated from many sources such as wind. Great contribution in terms of vibration energy is also generated by the city of Cairo, which is located closely to the pyramid of Khnum-Khufu and by the presence of the Nile river.

Given the large number of articles belonging to the previous literature, which has been conducted in the field of SAR, it is necessary to consider the following works: when the 'stop-and-go approximation', during the azimuth SAR focusing process, is no longer valid, it occurs that in addition to the constant Doppler frequency shift induced by the satellite's movement along its orbit, the target under consideration is subjected to micro-movement dynamics, such as mechanical vibrations or rotations. Such movements and accelerations, generated by the micro-movement dynamics, induce the micro-Doppler effect, which is directly attributable to various defocusing and spatial delocalization effects [59]. However, all those phenomena that generate artefacts from SAR image formation can represent an information resource. In fact, we exploit the artefact that is produced when a vibrating target is observed. The authors of [60] present an experimentally validated model that provides accurate localization and the shape of coupled echoes of vibrating targets in both near-field and far-field SAR images.

The successful experimental realization of polarimetric airborne SAR tomography is demonstrated for the first time in [61]. The authors present the concept of aperture synthesis for tomographic imaging for the case of a multi-baseline imaging geometry and discuss the constraints arising from the limited number of flight tracks. Superficial targets profiles detection and tomographic imaging connected to the electromagnetic penetration capabilities is achieved. This propriety depends on the used wavelengths, which is usually poor. The effects of phase miscalibration due to residual uncompensated atmospheric contribution and temporal SAR phase decorrelation is analyzed in [62]. The tomographic potential of high-resolution satellite images such as TerraSAR-X spotlight data of an urban environment is demonstrated in [63].

Biondi [49] proposed a new approach concerning the MM estimation of ships, processing the information given by sub-pixel tracking generated during the coregistration process of two re-synthesized time-domain and partially overlapped Doppler sub-apertures generated by splitting the raw data observed by a single wide azimuth band SAR image. Additionally, the predominant vibrational modes of different ships are then estimated in [50]. The performance analysis is conducted on one spotlight SAR image recorded by the COSMO-SkyMed satellite system, paving the way for application to the surveillance of land-based industry activities. A complete procedure for damage early-warning detection, by using MM estimation of critical sites, based on modal proprieties analysis has been assessed in [54]. Particularly, MM is processed to extract modal features such as natural frequencies and mode shapes generated by vibrations of large infrastructures. Problems connected to SAR acquisitions are that due to the poor penetrating action of electromagnetic waves within solid bodies, the ability to see through distributed targets is precluded. An imaging method based on the analysis of MMs present in volcanoes and generated by the Earth's underground seismic energy is described in [64], using penetrating tomographic imaging over a depth of about 3 km from the Earth's surface. The work describes also a complete assessment focused on showing the perfect space–time-frequency synchronization of SAR estimated vibrations with respect to seismic ripple estimated through in situ earthquake instruments.

Considering the experience gathered in the field of mechanical vibration estimation using radar, taking into account the imaging results addressed within volcanoes, considering all the calibration, validation as well as performance estimation experiments extensively described in [49,50,52,54], we processed several SAR images observed in the Vertical–Vertical

(VV) polarization, and the estimated MM allows us to visualize the principal internal components present in the pyramid.

We can state that the experimental results we propose definitively solve one of the oldest mysteries of human existence, the complete solution of the internal structure of Khnum-Khufu. To this end, in order to provide a more complete contribution to our work, we have firstly investigated the details of the external structure of all the pyramids belonging to the Giza Plateau (Khufu, Kefren and Menkaure); then, we concentrated on studying the internal structure of the pyramid of Khnum-Khufu alone, providing a complete and detailed 3D reconstruction of all the known and unknown chambers, based on tomographic SAR measurements. In the paper, we provide a complete list of the internal structures measured by tomography, each of them marked with a unique sequential number.

Table 1 lists the principal characteristics of the radar we used for the experiments.

**Table 1.** Principal parameters of the SAR acquisitions.

| SAR Parametrer | Value |
| --- | --- |
| Central frequency | 9.6 GHz |
| Chirp bandwidth | 400 MHz |
| Doppler bandwidth | 22 kHz |
| PRF | 2.0 kHz |
| PRT | 0.23 ms |
| Antenna length | 6 m |
| Type of acquisition | Spotlight |
| Polarization | HH, VV |
| Acquisition duration | 15 s |
| Platform velocity | 7 km/s |
| Observation height | 650,000 m |

## 2. Giza Plateau Presentation and Description

The pyramid of Khnum-Khufu is a monumental structure built mainly of granite blocks; its orientation is almost perfectly aligned to the north. The monumental complex of the Giza plateau is represented in Figure 1. The three pyramids, Khnum-Khufu (top right), Kefren (located in the center) and Menkaure (the last on the bottom left) can be observed. In this context, our work focuses on visualizing the vibrational tomographic profile of the pyramid of Khnum-Khufu. Figure 2a is the schematic representation of the north–south central section of the infrastructure. The figure represents the schematic of what is known, and the main parts of the infrastructure are numbered sequentially from 1 to 11. The object consists of the Zed and the King's chamber, with its sarcophagus inside. The Zed, the details of which can be seen in Figure 2b,c, is a large monument made entirely of granite, consisting of an upper roof made of two oblique granite slabs, and five parallel stone slabs, spaced at varying distances from each other. Each stone has its upper face not smooth, so each surface has a pronounced roughness. On the contrary, each of its lower faces is extremely smooth. Below this monument is the King's room. Both the Zed and the King's room are off-axis with respect to the apex of the pyramid and are located toward the south on the north–south symmetry plane. Object 2 of Figure 2a, is the Queen's room, a smaller volume object located on the axis of the pyramid and below the King's room. As can be seen from Figure 2a, Object 2, unlike the King's room, is located exactly under the apex of the pyramid. The last room is Object 3, which is also off-axis of the pyramid; it is also shifted to the south, but in this case, it is located underground. It is usually called the unfinished room. Object 4 represents a large corridor that connects the King's room with the Queen's room; it is called the Grand Gallery. Objects 5 and 6 are air ducts, while the remaining ducts 8, 9, and 10 connect the Grand Gallery with the Queen's room and the unfinished room (the one located below ground). Object 7 is the entrance to the pyramid, and finally, line 11 indicates the surface on which the pyramid sits.

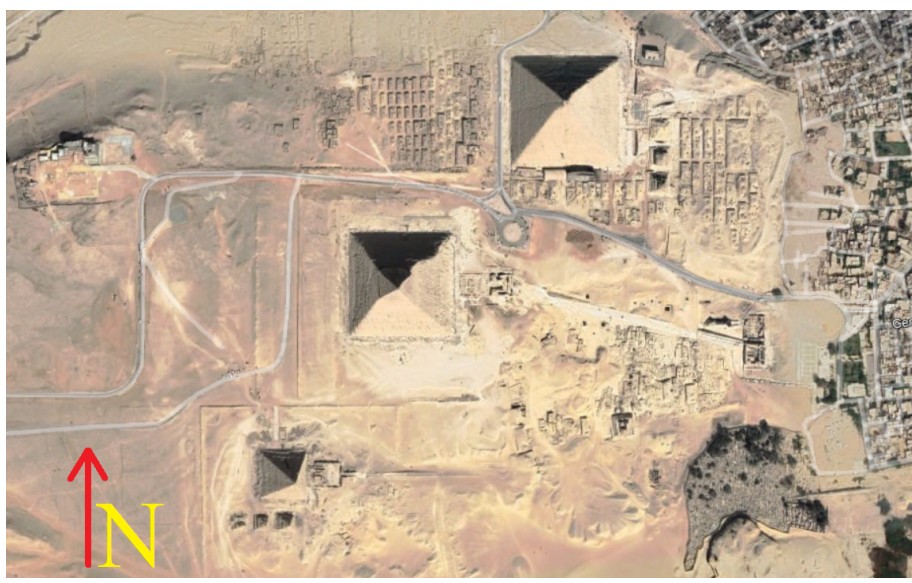

**Figure 1.** Optical satellite image of the Giza plateau. All pyramids are oriented to the North.

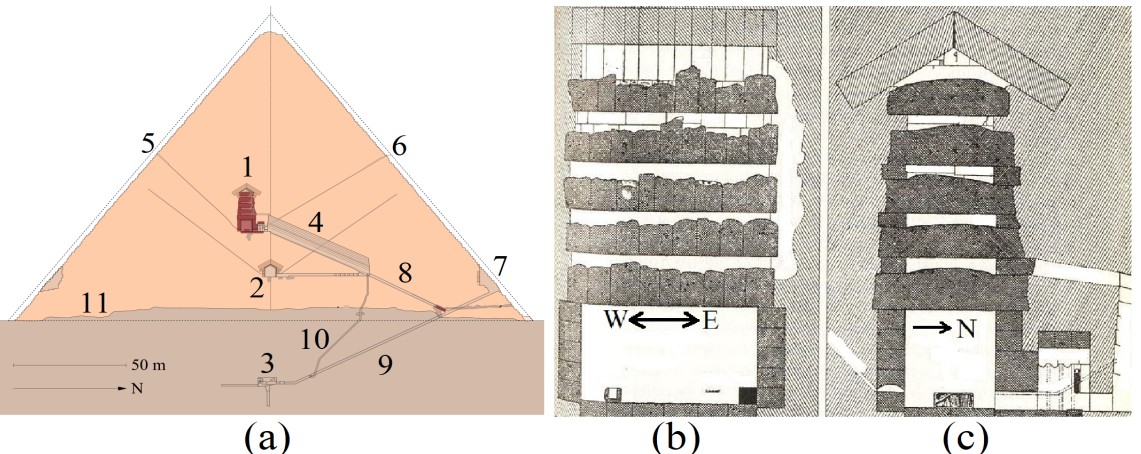

**Figure 2.** Khnum-Khufu diagram of the known interior structures. (**a**): East-side schematic view. (**b**): South-side view of the ZED. (**c**): East-side view of the ZED.

## 3. Methodology

In this work, the MM technique is used to perform sonic imaging by processing a single SAR image in the single-look-complex (SLC) configuration. The technique involves the MM estimation belonging to the Khnum-Khufu pyramid and is generated by the background ripple underground seismic activity that reflects superficial vibrations. The MM estimation is completed through MCA and performed in the Doppler direction. Multiple Doppler sub-apertures, SAR images with lower azimuth resolution, are generated to estimate the vibrational trend of some pixels of interest. The infra-chromatic displacement generated by Doppler centroid anomalies due to target motion and acceleration [65,66] is calculated through the pixel tracking technique, using high-performance sub-pixel coregistration [49, 50,52,54,64].

In Figure 3b, we present the SAR image of the pyramid. In order to show the temporal trend vibration, we examined a pixel located within the yellow circle 1. Figures 4–6 represent the time-domain displacement in magnitude, range and azimuth, respectively. The blue plot is the unfiltered temporal displacement trend, while the red function represents its positive envelope.

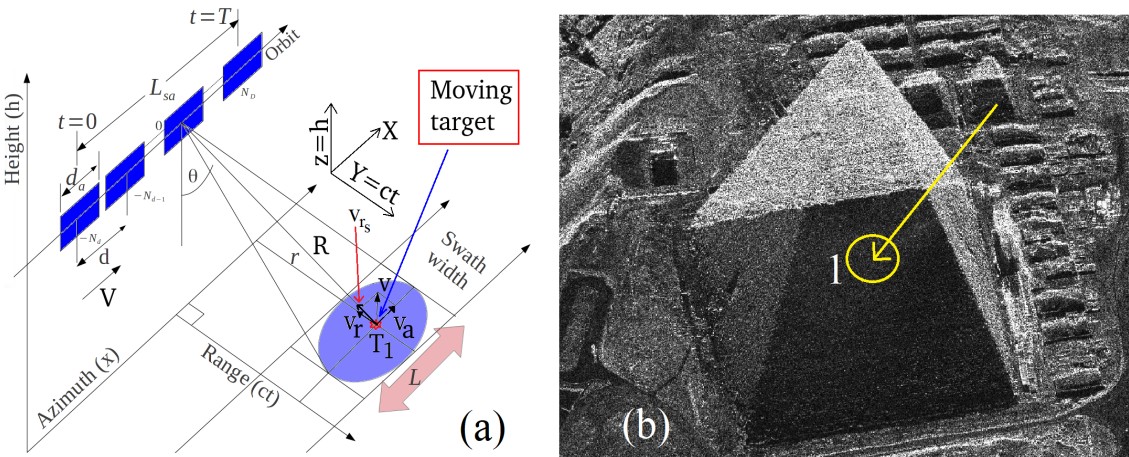

**Figure 3.** (**a**): SAR acquisition geometry. (**b**): SAR image in magnitude.

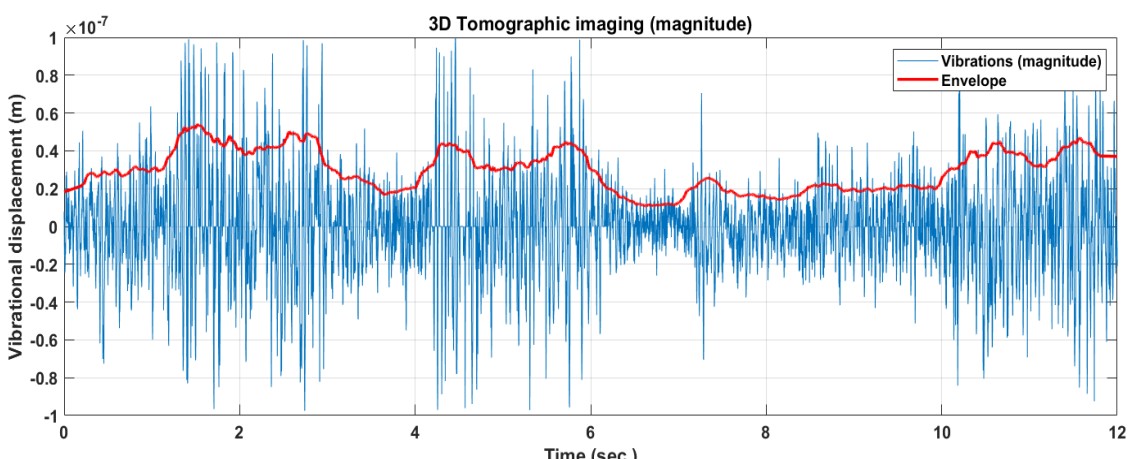

**Figure 4.** Time domain vibrations (magnitude).

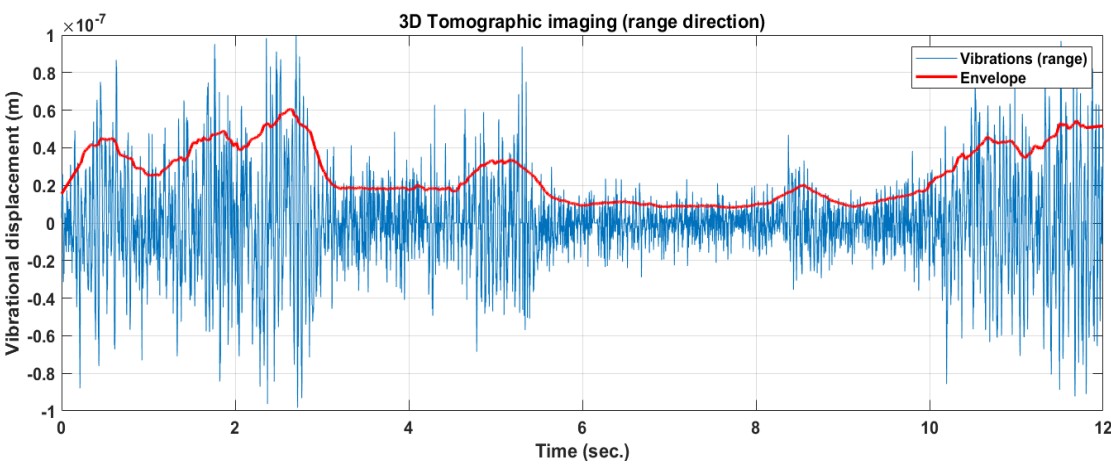

**Figure 5.** Time domain vibrations in the range direction (magnitude).

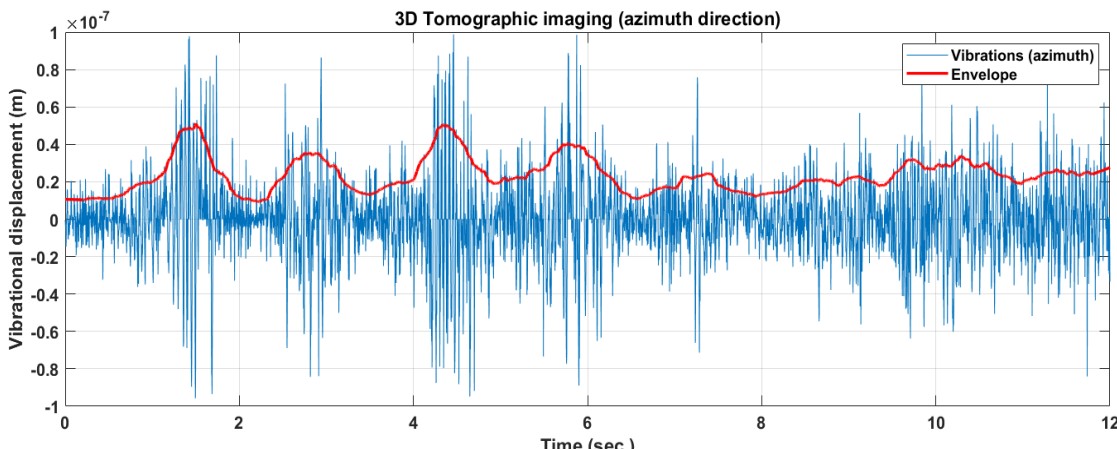

**Figure 6.** Time domain vibrations in the azimuth direction (magnitude).

Vibrations observed along the tomographic view direction, embedded into the multichromatic Doppler diversity, are focused along the height (or depth) dimension, developing metric resolution tomographic underground imaging.

The SAR synthesizes the electromagnetic image through a "side looking" acquisition, according to the observation geometry shown in Figure 3a, where:

- $r$ is the zero-Doppler distance (constant);
- $R$ is the slant-range;
- $R_0$ is the reference range at $t = 0$;
- $d_a$ is the physical antenna aperture length;
- $V$ is the platform velocity;
- $d$ is the distance between two range acquisitions;
- $L_{sa}$ is the total synthetic aperture length;
- $t$ is the acquisition time variable;
- $T$ is the observation duration;
- $t = 0$ and $t = T$ are the start and stop time acquisition, respectively;
- $L = \frac{\lambda r}{d_a}$ is the azimuth electromagnetic footprint width;
- $\theta$ is the incidence angle of the electromagnetic radiation pattern.

All the above parameters are related to the staring-spotlight SAR acquisition that is adopted in this work. The SAR data belonging to the electromagnetic image are formed through the focusing process that involves the application of a two-dimensional matched filter acting in the range direction and in the azimuth direction. The SLC signal resulting from compression is given by [67]:

$$s_{SLC}(k, x) = 2N\tau \exp\left[-j\frac{4\pi}{\lambda}r\right]\text{sinc}\left[\pi B_{c_r}\left(k - \frac{2R}{c}\right)\right]\text{sinc}[\pi B_{c_D}x] \tag{1}$$

$$\text{for } x = kt, \ k = \{0, 1, \dots, N-1\}, \ x = \{0, 1, \dots, M-1\}, \text{ with } N, M \in \mathbb{N}.$$

The focused SAR signal generated by the back-scattered electromagnetic energy of a point target supposed to be stationary is represented in (1). The terms $B_{c_r}$, and $B_{c_D} = \frac{4Nd}{\lambda r}$ are the total chirp and Doppler bandwidths respectively. The total synthetic aperture is equal to $L_{sa} = 2Nd$ and the azimuth resolution $\delta_D \approx \frac{1}{B_{c_D}} = \frac{\lambda R}{2L_{sa}}$. In (1) the $\frac{2\vec{R}}{c}$ term identifies the position in range where the maximum of the sinc function is located, while in azimuth it is centered around "zero". In the case where the peak of the sinc function has a nonzero coordinate along the azimuth dimension, Equation (1) can be recast as:

$$s_{SLC}(k, x) = 2N\tau \exp\left[-j\frac{4\pi r}{\lambda}\right]\text{sinc}\left[\pi B_{c_r}\left(k - L_{c_g}\right)\right]\text{sinc}\left[\pi B_{c_D}\left(x - L_{D_h}\right)\right] \tag{2}$$

$$\text{for } L_{c_g}, \ L_{D_h} \in \mathbb{N},$$

where the DFT is equal to:

$$
\begin{aligned}
S_{SLC_F}(n,q) &= DFT2\left\{2N\tau \exp\left[-j\frac{4\pi r}{\lambda}\right]\text{sinc}\left[\pi B_{c_r}\left(k-L_{c_g}\right)\right]\text{sinc}\left[\pi B_{c_D}\left(x-L_{D_h}\right)\right]\right\}\\
&= 2N\tau \exp\left[-j\frac{4\pi r}{\lambda}\right]\sum_{k=0}^{N-1}\sum_{x=0}^{M-1}\text{sinc}\left[\pi B_{c_r}\left(k-L_{c_g}\right)\right]\text{sinc}\left[\pi B_{c_D}\left(x-L_{D_h}\right)\right]\\
&\quad \exp\left(-j\frac{2\pi kn}{N}\right)\exp\left(-j\frac{2\pi xq}{M}\right)\\
&= 2N\tau \exp\left[-j\frac{4\pi r}{\lambda}\right]\frac{1}{\pi B_{c_r}}\text{rect}\left[\frac{n}{\pi B_{c_r}}\right]\frac{1}{\pi B_{c_D}}\text{rect}\left[\frac{q}{\pi B_{c_D}}\right]\\
&\quad \exp\left(-j2\pi nL_{c_g}\right)\exp\left(-j2\pi qL_{D_h}\right),
\end{aligned}
\tag{3}
$$

which has a rectangular shape.

### 3.1. Doppler Sub-Apertures Model

In this paper, we experiment a strategy that employs Doppler sub-apertures, which are generated to measure target motion. Figure 7 represents the used bandwidth allocation strategy. From the single SAR image, we calculate the 2D digital Fourier transform (DFT) which, according to (3), has a rectangular shape. As can be seen from Figure 7, $B_{C_D}$ is the total Doppler band synthesized with the SAR observation, while $B_{D_L} = \frac{B_{C_D}}{2}$ is the bandwidth we left out from the matched-filter boundaries to obtain a sufficient sensitivity to estimate target motions. In this context, formula (3) is the focused SAR spectrum, at maximum resolution, thus exploiting the whole band $\{B_{c_r}, B_{C_D}\}$; in accordance with the frequency allocation strategy shown in Figure 7, the following range-Doppler sub-apertures large matrix is constructed for the master multi-dimensional information:

$$
S_{SLC}(k,x)_M = \begin{bmatrix} S_{SLC}(k,x)_{M_{\{1,1\}}} & S_{SLC}(k,x)_{M_{\{1,2\}}} & S_{SLC}(k,x)_{M_{\{1,3\}}} & \dots & S_{SLC}(k,x)_{M_{\{1,N_D\}}} \end{bmatrix}
$$
$$
\text{for } N_D \in \mathbb{N},
\tag{4}
$$

and for the slave, the following large matrix is presented:

$$
S_{SLC}(k,x)_S = \begin{bmatrix} S_{SLC}(k,x)_{S_{\{1,1\}}} & S_{SLC}(k,x)_{S_{\{1,2\}}} & S_{SLC}(k,x)_{S_{\{1,3\}}} & \dots & S_{SLC}(k,x)_{S_{\{1,N_D\}}} \end{bmatrix}
$$
$$
\text{for } N_D \in \mathbb{N}.
\tag{5}
$$

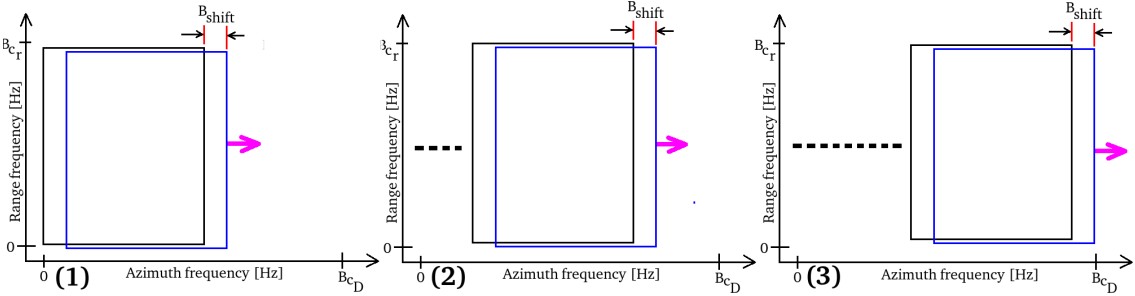

**Figure 7.** Doppler sub-aperture strategy. (**1**), (**2**), (**3**): Low, medium, high Doppler-frequency extrapolation respectively.

The explanation of the chirp-Doppler sub-aperture strategy, represented in Figure 7, is the following: We consider Figure 7, where master and slave sub-bands are generated by focusing the SAR image, where the matched filter is set to exploit a range-azimuth bandwidth equal to $B_{c_r}, B_{c_D} - B_{D_L}$. The not-processed bandwidths $B_{D_L}$ are divided into $N_D$ equally distributed bandwidths steps, respectively. At this point, $N_c$ rigid shifts of the master–slave system are made along the azimuth bandwidth domain; this is made to

populate the entire row of (4) and (5). The process is repeated $N_D$ times for each shift in azimuth; in fact, Figure 7(1–3) represent the azimuth frequency variation strategy when the Doppler bandwidth is located at $N_D$. At each Doppler frequency shift $\frac{B_{c_D} - B_{D_L}}{N_D}$, every element of (4) and (5) is populated.

### 3.2. Doppler Sub-Aperture Strategy

The decomposition of the SAR data into Doppler sub-apertures is formalized in this subsection, which is performed starting from the spectral representation of the focused SAR data. To this end, notice that the generic *i*-th chirp sub-aperture two-dimensional DFT of (2) is given by:

$$
\begin{aligned}
S_{SLC_{F_i}}(n,q) &= DFT2\left\{ 2N\tau \exp\left[-\jmath\frac{4\pi r}{\lambda}\right] \mathrm{sinc}\left[\pi B_{c_{r_i}}\left(k - L_{c_g}\right)\right] \mathrm{sinc}\left[\pi B_{c_D}\left(x - L_{D_h}\right)\right] \right\} \\
&= 2N\tau \exp\left[-\jmath\frac{4\pi r}{\lambda}\right] \sum_{k=0}^{N-1}\sum_{x=0}^{M-1} \mathrm{sinc}\left[\pi B_{c_{r_i}}\left(n - L_{c_g}\right)\right] \mathrm{sinc}\left[\pi B_{c_D}\left(q - L_{D_h}\right)\right] \\
&\quad \exp\left(-\jmath\frac{2\pi kn}{N}\right) \exp\left(-\jmath\frac{2\pi xq}{M}\right) \\
&= 2N\tau \exp\left[-\jmath\frac{4\pi r}{\lambda}\right] \frac{1}{\pi B_{c_{r_i}}} \mathrm{rect}\left[\frac{n}{\pi B_{c_{r_i}}}\right] \frac{1}{\pi B_{c_D}} \mathrm{rect}\left[\frac{q}{\pi B_{c_D}}\right] \exp\left(-\jmath 2\pi n L_{c_g}\right) \exp\left(-\jmath 2\pi q L_{D_h}\right).
\end{aligned}
\tag{6}
$$

From the last equation, it turns out that a single point stationary target has a two-dimensional rectangular nature with total length proportional to the range-azimuth bandwidths, respectively. The phase term $\exp\left(-\jmath 2\pi n L_{c_g}\right) \exp\left(-\jmath 2\pi q L_{D_h}\right)$ is due to the sinc function dislocation in range and azimuth when the SLC SAR data are considered. In the SAR, the movement of a point target with velocity in both the range and azimuth direction is immediately warned by the focusing process, resulting in the following anomalies:

- Azimuth displacement in the presence of target constant range velocity;
- Azimuth smearing in the presence of target azimuth velocity or target range accelerations;
- Range-walking phenomenon, visible as range defocusing, in the presence of target range speed; backscattered energy can be detected over one or more range resolution cells.

According to (6), it turns out that a single point stationary target has a two-dimensional rectangular nature with total length proportional to the range-azimuth bandwidths respectively. The phase term $\exp\left(-\jmath 2\pi n L_{c_g}\right) \exp\left(-\jmath 2\pi q L_{D_h}\right)$ is due to the sinc function dislocation in range and azimuth when the SLC SAR data are considered. In the SAR, the movement of a point target with velocity in both range and azimuth direction is immediately warned by the focusing process, resulting in the following anomalies:

- azimuth displacement in the presence of target constant range velocity;
- azimuth smearing in the presence of target azimuth velocity or target range accelerations;
- range-walking phenomenon, visible as range defocusing, in the presence of target range speed, backscattered energy can be detected over one or more range resolution cells.

In real cases, the backscattered energy from moving targets is distributed over several range-azimuth resolution cells. As a matter of fact, considering the point-like target $T_1$ (of Figure 3a) that is moving at $\vec{v}_t$ whose (slant-range)-azimuth and acceleration components are $\{v_{r_s}, v_a\}$, and $\{a_r, a_a\}$, respectively, then we can write

$$R^2(t) = (Vt - S_a)^2 + (R_0 - S_r)^2 \text{ with } S_r = v_{r_s}t + \frac{1}{2}a_r t^2 \text{ and } S_a = v_a t + \frac{1}{2}a_a t^2$$

$$|R(t)| = |R_0 - S_r| \left\{ 1 + \frac{(Vt - S_a)^2}{(R_0 - S_r)^2} \right\}^{\frac{1}{2}}. \tag{7}$$

We consider the following Taylor expansion:

$$(1 + x)^\beta \approx 1 + \beta x \tag{8}$$

and that $R_0 - S_r \approx R_0$, and $(Vt - S_a)^2 \approx V^2 t^2 - 2VtS_a$, (7) can be written in the following form:

$$|R(t)| = \left\{ |R_0 - S_r| + \frac{1}{2} \frac{(Vt - S_a)^2}{(R_0 - S_r)} \right\} = |R_0 - S_r| + \frac{V^2 t^2}{2R_0} \left( 1 - \frac{2S_a}{Vt} \right) \tag{9}$$

$$= R_0 - S_r + \frac{V^2 t^2}{2R_0} - \frac{VtS_a}{R_0}$$

$$= R_0 - v_{r_s}t - \frac{1}{2}a_r t^2 + \frac{V^2 t^2}{2R_0} - \frac{Vt\left(v_a t + \frac{1}{2}a_a t^2\right)}{R_0} \tag{10}$$

$$= R_0 - v_{r_s}t - \frac{1}{2}a_r t^2 + \frac{V^2 t^2}{2R_0} - \frac{Vv_a t^2}{R_0} - \frac{Va_a t^3}{2R_0}.$$

Neglecting $\frac{Va_a t^3}{2R_0}$ and approximating $(V^2 - 2Vv_a) \approx (V - v_a)^2$, Equation (10) can be written like:

$$|R(Vt)| = R_0 - v_{r_s}t + \frac{t^2}{2R_0}\left[(V - v_a)^2 - R_0 a_r\right], \tag{11}$$

recasting (11) in terms of $x = Vt$, we obtain [65]:

$$|R(x)| = R_0 - \epsilon_{r_1} x + \left[(1 - \epsilon_{c_1})^2 - \epsilon_{r_2}\right]\frac{x^2}{2R_0}, \quad x = Vt, \tag{12}$$

where:

- $\epsilon_{r_1} = \frac{v_{r_s}}{V}$ (due to range velocity);
- $\epsilon_{r_2} = \frac{a_r R_0}{V^2}$ (due to range acceleration);
- $\epsilon_{c_1} = \frac{v_c}{V}$ (due to azimuth velocity).

The above presented terms, as shown in [65], modify the received signal, and should be taken into account in Equation (6) .

## 4. Tomographic Model

Considering a single SLC image from which we applied the MCA according to the frequency allocation strategy depicted in Figure 7, the tomogram represented by the line of contiguous pixels shown in Figure 8 is calculated. The vibrations present on the tomographic plane extending from the Earth's surface to a depth of a few kilometers is assessed. The figure represents a series of harmonic oscillators anchored on each pixel of the tomographic line, which is symbolically represented as a spring linked to a mass and oscillating due to the application of harmonic vibrations. Each wave generated by each harmonic oscillator bounces off the surface of the Earth, as there is an abrupt variation in the density of the medium (the ground-air boundary). On each pixel, a vibrational phasor is observed in time applying Doppler MCA [52,54]. Through the orbital change of view

(which is performed in azimuth), an effective subsurface in-depth vibrational scan of the Earth is achieved.

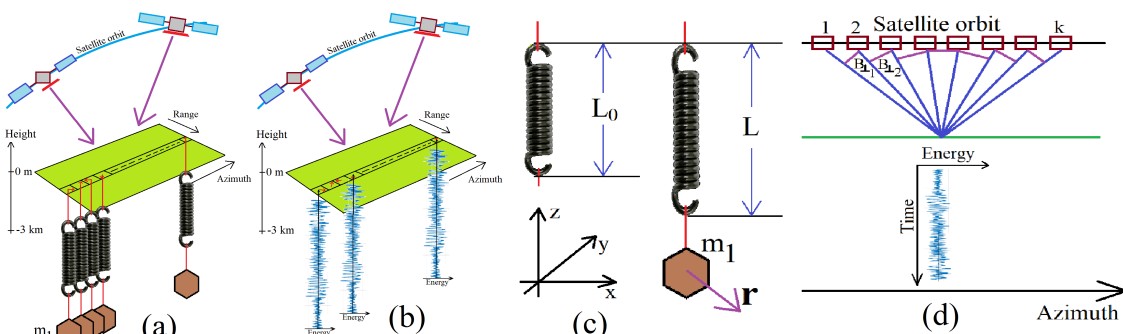

**Figure 8.** Tomographic acquisition geometry. (**a**): SAR acquisition of targets represented by harmonic mass oscillators. (**b**): SAR acquisition of targets represented by time-domain harmonic vibration functions. (**c**): Static representation in a 3D spatial reference system of a spring when expanded by a mass. The spring from a length $L_0$ is dilated by gravity acting on the mass $m_1$, to a length $L$. (**d**): Details of the tomographic acquisition geometry along the satellite orbit. All orthogonal baselines are represented.

*Vibrational Model of the Earth*

The proposed vibrational model of the Earth surface is schematically shown in Figure 8a,b. The geometrical reference system for both sub-pictures is the range, azimuth and altitude three-dimensional space. For the present case, the vertical dimension represents the depth below the topographic level (for this specific case, the medium boundary is represented by the green plane). The tomographic line of interest is constituted of the series of contiguous pixels laying on the green plane. As can be seen from Figure 8a, on each pixel belonging to the tomographic line, a mass is hanging using a spring. This system is now induced to oscillate harmonically due to the seismic background ripple, the city of Cairo human activities and the wind force field. These oscillations are schematized as the vibration energy function visible in Figure 8b. In this context, the radar instantaneously perceives this coherent harmonic oscillation. From a mathematical point of view, the Earth displacement is perceived as a complex shift belonging on each pixel of interest. Each instantaneous displacement is estimated between the master image with respect to the slave, where the shifts are estimated through the pixel tracking technique [52,54]. The computational flowchart in Figure 9 schematically describes the computational flow from a single SAR image to a tomographic estimate. Computational block 1 represents the single SAR image in the SLC configuration. Computational blocks 2 and 3 are bandpass filters and operate according to the frequency allocation strategy in Figure 7. These filters play a crucial role in the estimation of displacements that are equal to vibration. Computational block 4 represents a spatial filter that only performs vibration estimation on the points representing the tomographic line under consideration. Computational block 5 implements the vibration metric algorithm, and finally, computational block 6 performs the focusing of the estimated acoustic waves in order to obtain the internal tomography of the distributed target crossed by the tomographic line.

The number of tomographic independent looks (depending on the total number of Doppler sub-apertures) is defined by the parameter $k$.

We suppose now the spring is being perturbed by an impulsed force. According to this perturbation, the rope begins to vibrate, describing a harmonic motion (in this context, we are not considering any form of friction). The resulting perturbation moves the rope through the space–time in the form of a sinusoidal function. The seismic wave will then reach a constraint end that will cause it to reflect in the opposite direction. The reflected wave will then reach the opposite constraint that will make it reflect in the original direction and return in the initial location, maintaining the same frequency and amplitude. According

to Classical Physics principles, the rebounding wave is superimposed on the arriving wave, and the interference of two sine waves with the same amplitude and frequency propagating in opposite directions leads to the generation of an ideal and perpetual standing wave on the spring. Each vibrational channel is now considered when the spring is able to oscillate into the three-dimensional space, according to specific perturbation nature. When the Earth vibrates, it happens that the length of the spring must also fluctuate. This phenomenon causes oscillations in the tension domain of the spring. It is clear that these oscillations (i.e., the longitudinal ones) propagate through a frequency approximately twice as high as the frequency value of the transverse vibrations. The coupling between the transverse and longitudinal oscillations of the spring can essentially be modeled through non-linear phenomena.

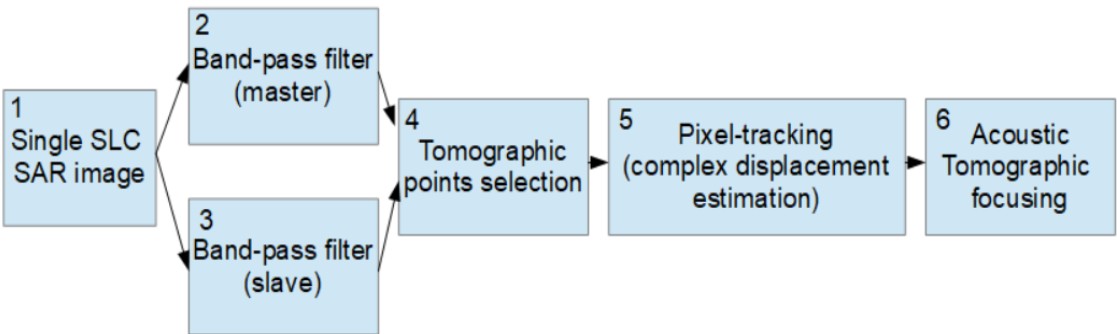

**Figure 9.** Computational flowchart for estimating the internal results.

Figure 8c,d illustrates the oscillating model in the Euclidean space–time coordinates (x,y,z,t), where the satellite motion has been purified from any orbital distortions, so that the geometric parameters used to perform the tomographic focusing can be rigorously understood. From Figure 8c, $L$ is the length of the spring when it is at its maximum tension, while $L_0$ is its length when no mass is present. Finally, the spring has been considered to have an elastic constant equal to $\xi$. The vibrational force applied to the mass $m1$ of Figure 8c is equal to [68]:

$$F = -4\xi \mathbf{r}\left(1 - \frac{L_0}{\sqrt{L^2 + 4\mathbf{r}^2}}\right). \tag{13}$$

If $\mathbf{r} \ll L$, (13) is expanded in the following series:

$$F = -4\xi \mathbf{r}(L - L_0)\left(\frac{\mathbf{r}}{L}\right) - 8\xi L_0\left[\left(\frac{\mathbf{r}}{L}\right)^3 - \left(\frac{\mathbf{r}}{L}\right)^5 + \dots\right], \tag{14}$$

where a precise approximation of (14) is the following cubic restoring force:

$$F = m\ddot{\mathbf{r}} \approx -4\xi \mathbf{r}(L - L_0)\left(\frac{\mathbf{r}}{L}\right)\left[1 + \frac{2L_0}{(L - L_0)}\left(\frac{\mathbf{r}}{L}\right)^2\right]. \tag{15}$$

where the non-linearity condition dominates when $L \approx L_0$. If we define:

$$\omega_0 = \frac{4\xi}{m}\left[\frac{(L - L_0)}{L}\right], \tag{16}$$

and

$$\xi = \frac{2L_0}{L^2}(L - L_0), \tag{17}$$

and taking into account (15) we have:

$$\ddot{\mathbf{r}} + \omega^2 \mathbf{r}\left(1 + \zeta \mathbf{r}^2\right) = 0. \tag{18}$$

When considering damping and forcing, (18) is modified as:

$$\ddot{\mathbf{r}} + \lambda \dot{\mathbf{r}} + \omega^2\left(1 + \zeta \mathbf{r}^2\right)\mathbf{r} = \mathbf{f}(\omega t), \tag{19}$$

where $\mathbf{f}(\omega t)$ is the forcing term and $\lambda$ is the damping coefficient. If non-linearity of (19) is sufficiently low, it can be reduced into the following two-degree-of-freedom linear harmonic oscillator:

$$\mathbf{r}(t) = (a\cos\omega_0 t, b\sin\omega_0 t)\exp\left(\frac{-\lambda t}{2}\right) \tag{20}$$

where $\{a, b\}$ are the instantaneous shifts estimated by the coregistrator. The harmonic oscillator (20) contains the displacement parameters $\epsilon_{r_1}, \epsilon_{r_2}, \epsilon_{c_1}$ estimated by (12). According to Figure 8d the vector representation of $k$ samples of the time-domain function (20) consisting in the following multi-frequency data input is considered:

$$\mathbf{Y} = [\mathbf{y}(1), \dots, \mathbf{y}(k)], \in \mathbf{C}^{k \times 1}. \tag{21}$$

We introduce here the sonic steering matrix $\mathbf{A}(z) = [\mathbf{a}(z_{min}), \dots, \mathbf{a}(z_{MAX})], \in \mathbf{C}^{k \times F}$ contains the phase information of to the Doppler frequency variation of the sub-aperture strategy, associated to a source located at the elevation position $\mathbf{z} \in \{z_{min}, z_{MAX}\}$,

$$\mathbf{A}(\mathbf{K}_z, \mathbf{z}) = \begin{bmatrix} 1, \exp(j2\pi k_{z_2} t z_0), \dots, \exp(j2\pi k_{z_{k-1}} t z_0) \\ 1, \exp(j2\pi k_{z_2} t z_1), \dots, \exp(j2\pi k_{z_{k-1}} t z_1) \\ \dots \\ 1, \exp(j2\pi k_{z_2} t z_{F-1}), \dots, \exp(j2\pi k_{z_{k-1}} t z_{F-1}) \end{bmatrix}, \tag{22}$$

where $\mathbf{K}_z = \frac{4\pi B_\perp}{\lambda \mathbf{r}_i \sin\theta}$, $i = 1, \dots, k$, $B_\perp$ is the $i$-th orthogonal baseline which is visible in Figure 8d, and $\mathbf{r}_i$ is the $i$-th slant–range distance. The standard sonic tomographic model is given by the following relation:

$$\mathbf{Y} = \mathbf{A}(\mathbf{K}_z, \mathbf{z})\mathbf{h}(\mathbf{z}), \tag{23}$$

where in (23), $\mathbf{h}(\mathbf{z}) \in \mathbf{C}^{1 \times F}$, inverting (23), I finally find the following tomographic solution:

$$\mathbf{h}(\mathbf{z}) = \mathbf{A}(\mathbf{K}_z, \mathbf{z})^\dagger \mathbf{Y}. \tag{24}$$

In (24), the steering matrix $\mathbf{A}(\mathbf{K}_z, \mathbf{z})$ represents the best approximation of a matrix operator performing the digital Fourier transform (DFT) of $\mathbf{Y}$. The tomographic image $\mathbf{h}(\mathbf{z})$, which represents the spectrum of $\mathbf{A}(\mathbf{K}_z, \mathbf{z})$, is obtained by doing pulse compression.

The tomographic resolution is equal to $\delta_T = \frac{\lambda R}{2A}$, where $\lambda$ is the sound wavelength over the Earth, $R$ is the slant range, and $A$ is the orbit aperture considered in the tomographic synthesis; in other words, it consists of the Doppler bandwidth used to synthesize the sub-apertures. The maximum tomographic resolution obtainable using these SLC data, synthesized at 24 kHz, is as follows. Considering an average speed of propagation of the seismic waves of about $v \approx 6000$ m/s, a frequency of investigation set by us equal to 12,500 Hz, the wavelength of these vibrations is equal to about $\lambda = \frac{v}{f} \approx \frac{6000}{25,000} \approx 0.24$ m. Considering the above parameters, extending the tomography to the maximum orbital aperture equal to half the total length of the orbit, therefore about 42,000 m, with $R = 650,000$ m, the tomographic resolution is equal to $\delta_z = \frac{\lambda R}{2A} = \frac{0.24 \cdot 650,000}{2 \cdot 84,000} \approx 0.92$ m. This is the tomographic resolution set to calculate all the experimental parts shown in Section 5.

## 5. Experimental Results

In this section, we will show all the experimental results that have been made and have been divided into external, and internal, experimental results. In the first case, we show the results provided by SAR interferometry (InSAR). These consist of the evaluation of radar interferometric fringes to demonstrate the actual shapes of the outer facades of all pyramids in the Giza Plateau. The detailed explanation is provided in Section 5.1. Similarly, the results for the internal vibrational tomography analysis of Khnum-Khufu alone are discussed in detail in Section 5.2.

### 5.1. External Experimental Results

This subsection shows and discusses all the results obtained with the aim of revealing new features of the external appearance of all pyramids residing on the Giza Plateau. In order to achieve this goal, we employed the InSAR technique and evaluated the nature of the interferometric fringes and discovered, through the measurement of their inclination, that all pyramids do not each consist of four faces but of eight faces. We processed data according to the images listed in Table 2, from picture 1 to 6. We found that each face of each pyramid had an inwards bow that became more relevant closer to the ground much like a trough. Figure 10 shows the interferometric fringes generated by two SAR repeat-pass acquisitions with a suitable spatial baseline in order to generate a series of well-estimated interferometric fringes imprinted on the faces of the pyramids. The interferometric acquisition was performed along a time baseline equal to the complete orbital cycle of the single COSMO-SkyMed Second Generation (CSG) satellite, which coincides with 16 days. In spite of the substantial number of waiting days, the interferometric acquisition appears not to be very noisy, and this quality is confirmed through the evaluation of the coherence parameter whose map is represented in Figure 11. This result appears very good, as a large part of the figure, removing all the areas where the radar shadow is present, and it maintains coherence levels very close to 1.

**Table 2.** List of processed COSMO-Sky-Med SAR images.

| Picture | Date | Orbit | Beam | Polarization | Experiment |
|---|---|---|---|---|---|
| 1 | 28 October 2021 | Right-descending | 06 (master) | HH | External |
| 2 | 13 November 2021 | Right-descending | 06 (slave) | HH | External |
| 3 | 27 October 2021 | Right-descending | 08 (master) | HH | External |
| 4 | 12 November 2021 | Right-descending | 08 (slave) | HH | External |
| 5 | 24 July 2021 | Right-ascending | 39 (master) | HH | External |
| 6 | 9 August 2021 | Right-ascending | 39 (slave) | HH | External |
| 7 | 25 February 2022 | Left-descending | 46 (single image) | VV | Internal |
| 8 | 16 November 2021 | Right-descending | 48 (single image) | HH | Internal |
| 9 | 22 February 2022 | Right-descending | 48 (single image) | VV | Internal |
| 10 | 16 February 2022 | Right-ascending | 48 (single image) | VV | Internal |
| 11 | 25 March 2022 | Right-descending | 48 (single image) | VV | Internal |
| 12 | 26 April 2022 | Right-descending | 48 (single image) | VV | Internal |

Figure 12a represents the details of the InSAR fringes measured on the three pyramids (Khufu, Kefren, and Menkaure), while Figure 12b shows the detail of the pyramid of Khnum-Khufu, and finally, Figure 12c is the pyramid of Kefren. Figure 13a–d show the particular representations of the SAR interferometric fringes observed on the east face of the Pyramid of Khnum-Khufu (in box (a)). The remaining boxes (b), (c) and (d), represent the first, second and third interferometric fringes observed on the north face of the same pyramid, starting from the bottom. The inclination of the entire face of the pyramid is clearly observed, having symmetry along the height of the geometric figure. The north face of the Khnum-Khufu pyramid is depicted in Figure 14a, while the details of the first, second and third fringe (starting from the ground plane) are shown in Figure 14b–d, respectively. Here again, the same effect is observed whereby the single face is divided into two indented

half-faces. The west face also presents the same architectural feature; in fact, Figure 15a depicts the extension of the interferometric fringes extended over the entire south face, while the details of the first, second and third fringes (always starting from the ground plane) are shown in Figure 15b–d, respectively.

We now move to the pyramid of Kefren to repeat the same experiments where the same qualitative evaluation of the SAR interferometric fringes present on the three facets of it, except for the south one, will be carried out, reaching the identical conclusions made for the pyramid of Khufu. Figure 16a–d shows the particular representation of the SAR interferometric fringes observed on the west face of the Pyramid of Kefren (in box (a)). The remaining boxes (b), (c) and (d), represent the first, second and third interferometric fringes observed on the north face of the same pyramid, starting from the bottom. The north front of the second Kefren pyramid is analyzed in Figure 17a, depicting the extent of the interferometric fringes as a whole, while the details of the first, second and third fringes (starting from the ground plane) are shown in Figure 17b–d, respectively. The south façade is also studied in Figure 18a, along with its first, second and third fringe details (starting from the ground plane), which are shown in Figure 18b–d, respectively.

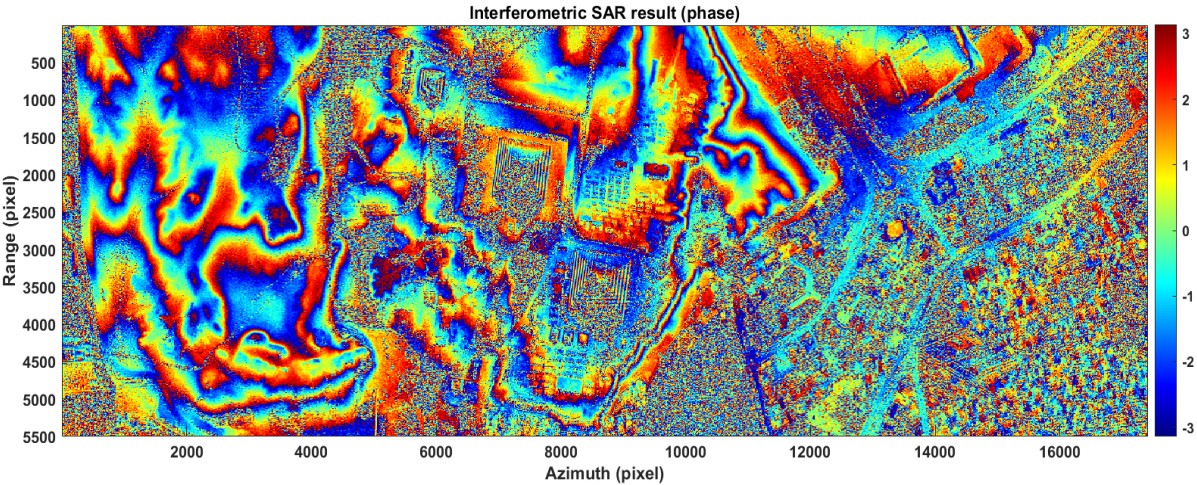

**Figure 10.** SAR image representing the interferometric repeat-pass phase of the Giza plateau.

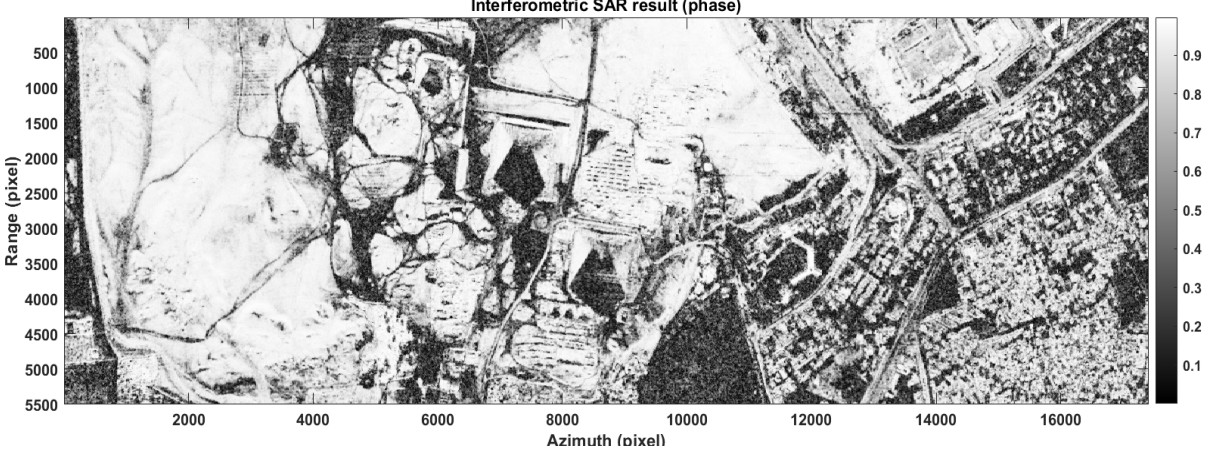

**Figure 11.** SAR image representing the interferometric repeat-pass coherence of the Giza plateau.

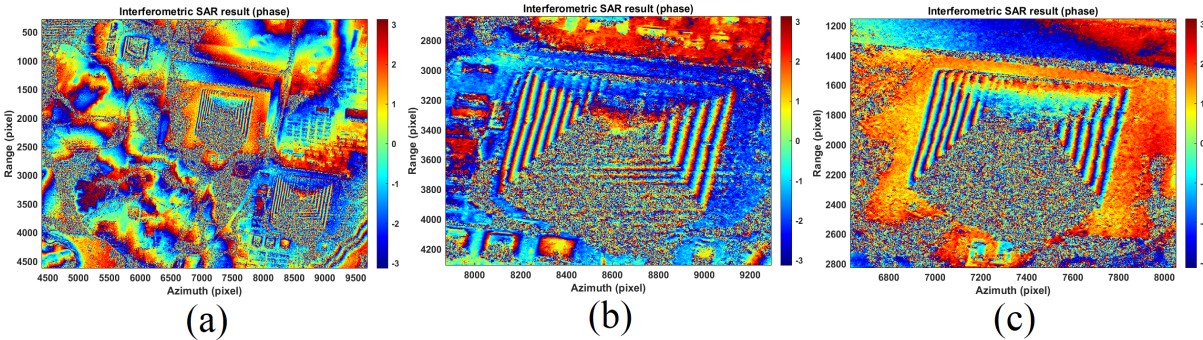

**Figure 12.** Interferometric SAR results. (**a**): Interferometric fringes of the Giza plateau. (**b**): Interferometric fringes of Khnum-Khufu. (**c**): Interferometric fringes of Khefren.

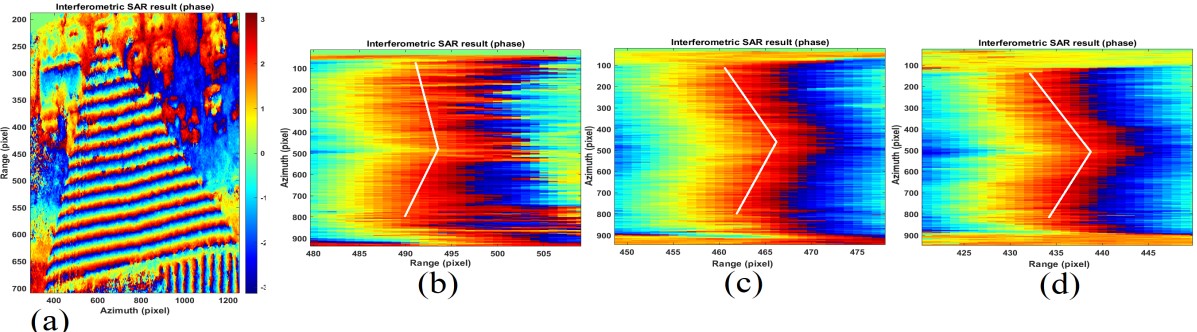

**Figure 13.** Representation in particular of the pyramid of Khnum-Khufu. Interferometric phases. (**a**): east side. (**b**–**d**): magnification of interferometric fringe 1, 2, and 3.

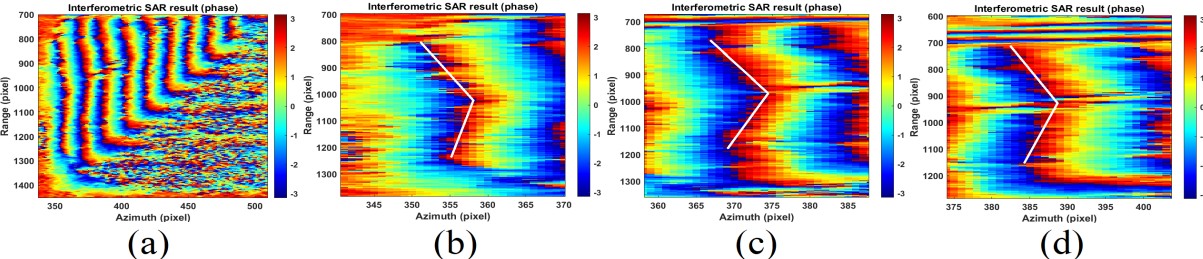

**Figure 14.** Representation in particular of the pyramid of Khnum-Khufu. Interferometric phases. (**a**): north side. (**b**–**d**): magnification of interferometric fringe 1, 2, and 3.

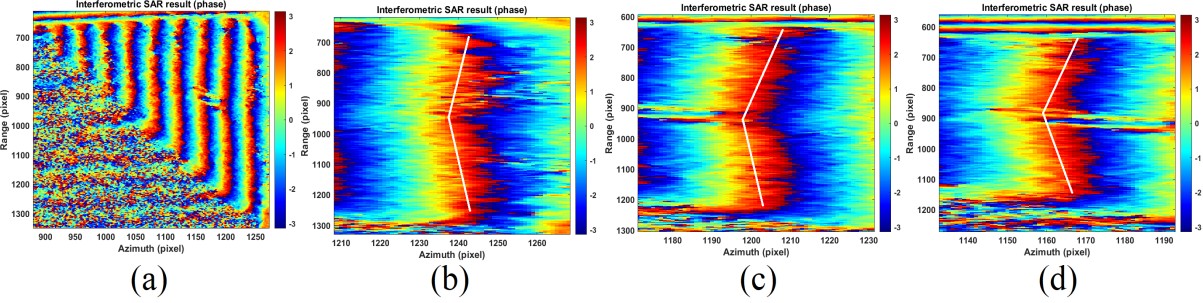

**Figure 15.** Representation in particular of the pyramid of Khnum-Khufu. Interferometric phases. (**a**): south side. (**b**–**d**): magnification of interferometric fringe 1, 2, and 3.

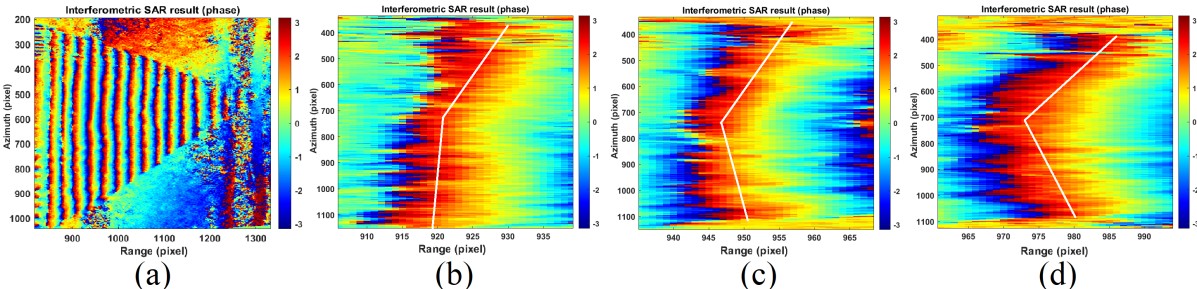

**Figure 16.** Representation in particular of the pyramid of Kefren. Interferometric phases. (**a**): west side. (**b–d**): magnification of interferometric fringe 1, 2, and 3.

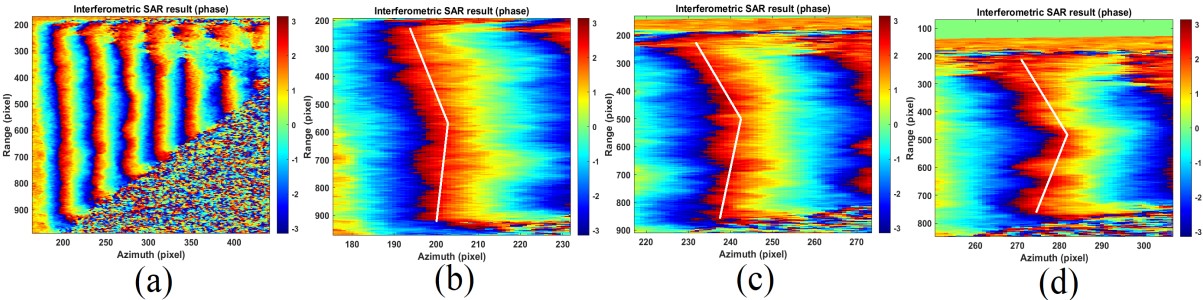

**Figure 17.** Representation in particular of the pyramid of Kefren. Interferometric phases. (**a**): north side. (**b–d**): magnification of interferometric fringe 1, 2, and 3.

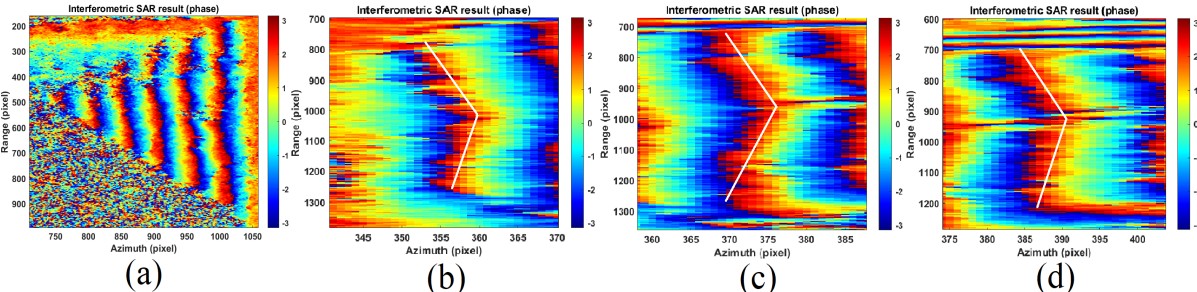

**Figure 18.** Representation in particular of the pyramid of Kefren. Interferometric phases. (**a**): south side. (**b–d**): magnification of interferometric fringe 1, 2, and 3.

Although much smaller, the pyramid of Menkaure is also well represented by the interferometric radar. As a matter of fact, Figure 19a–d depict the entire pyramid (in Figure 19a), while Figure 19b–d depict the first fringe starting from the ground plane of the east, west, and north faces, respectively. Surprisingly, the pyramid of Menkaure also consists of eight facets and not four (while maintaining the fact that the south face could not be observed because it was in radar shadow). This subsection, which focused on presenting the results of external measurements alone, ends by rigorously demonstrating, through radar measurements, the eight-sided nature of the three pyramids of Khufu, Kefren and Menkaure. In the next subsection, the internal measurements that were made from space will be detailed in order to carry out for the first time the complete internal mapping of all structures belonging to the pyramid of Khnum-Khufu alone. In order to ensure the reliability of the data, we show the correspondence of the displacement of the anomaly trend on the west face only, where we compared the data measured by the radar with those generated by the Lidar. Here, we show the plots of both profiles with relative errors. In Figure 20a, we show the displacement trend of the interferometric fringe in Figure 16c. The unfiltered trend is represented by the blue function, while the filtered trend is represented by the black trajectory. Figure 20b instead shows the trend of the measurements, taken along the same trajectory travelled by the interferometric fringes,

but in this case generated by the Lidar. Again, the unfiltered trend is represented by the blue function, while the filtered trend is represented by the black trajectory. Figure 20c shows the superposition of SAR and Lidar measurements, confirming the same trend. In fact, Figure 21a shows the superposition of the filtered functions of both SAR and Lidar, and Figure 21b quantifies the error existing between the two. Again, the blue-colored function is the unfiltered one, while the black one represents the filtered. As can be seen from the trend, the error is very low: it is quantifiable at about 0.1 meters at its maximum value of about 35 cm.

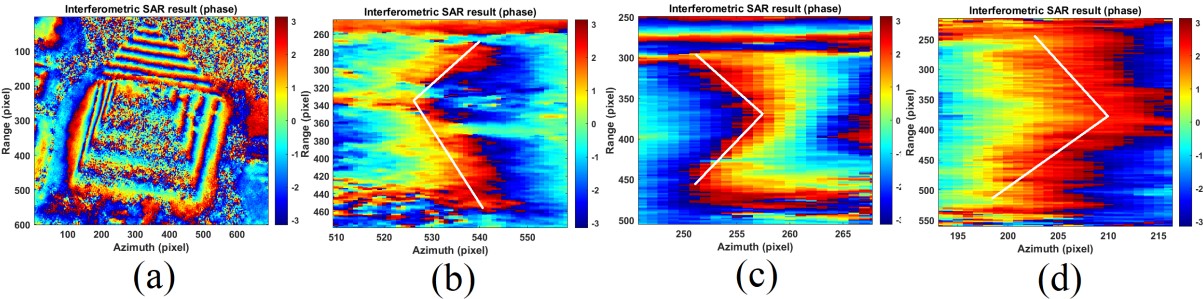

**Figure 19.** Representation in particular of the pyramid of Menkaure. Interferometric phases. (**a**): Menkaure pyramid. (**b**): east side interferometric fringe particular. (**c**): west side interferometric fringe particular. (**d**): north side interferometric fringe particular.

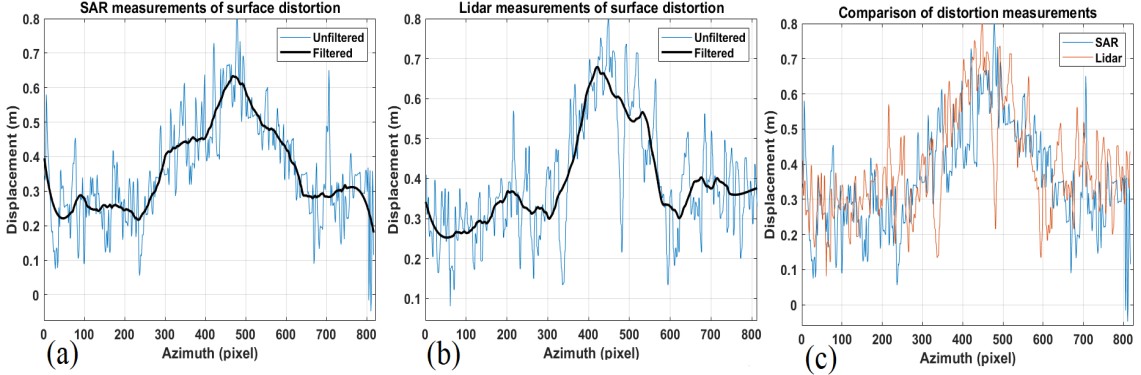

**Figure 20.** Displacement measurements. (**a**): SAR displacement. (**b**): Lidar displacement. For (**a**,**b**), the blue functions are the unfiltered trends, while the black plot are filtered. (**c**): Comparison of SAR versus Lidar displacement measurements.

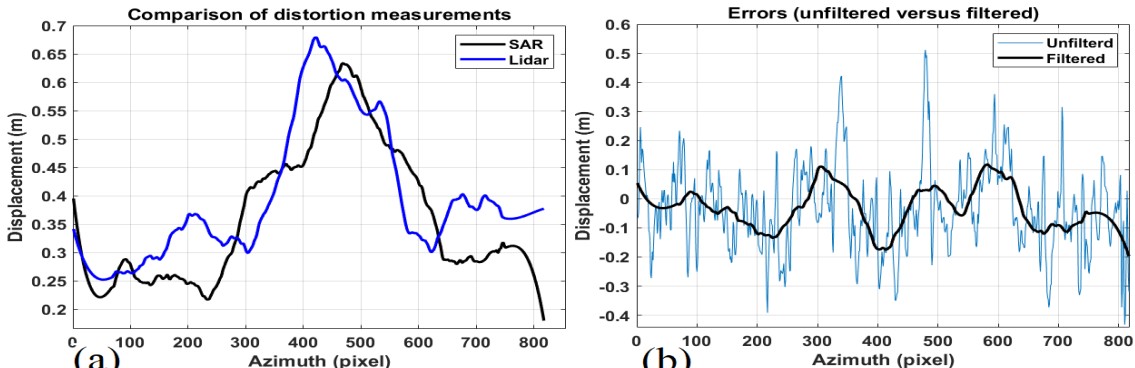

**Figure 21.** (**a**): Filtered displacement measurements. (**b**): Errors between SAR and Lidar displacement trends.

## 5.2. Internal Experimental Results of Khnum-Khufu

Data analysis obtained using the SAR tomographic Doppler imaging technique was able to provide clear objective elements to understand the internal structure of the pyramid

of Khnum-Khufu. When proposing our results, we start by describing the well-known structures and then move on to the description of some of the unknown structures. The internal imaging obtained from multiple angles allows us to obtain an accurate 3D model that gave us the possibility, like never before, of taking a look inside one of the most important and mysterious megalithic monuments in the world. We processed data according to the images listed in Table 2, from pictures 7 to 12. In the calculation of the internal experimental results, we have not taken into account the effects due to multiple reflections (multipath) of the sound wave. In fact, as this research is in its current state, we will also try to take these interference effects into account in the near future, as they could negatively perturb the tomographic measurement. One fundamental thing, however, has been to greatly attenuate disturbances due to both radar electromagnetic multipath and disturbances due to SAR layover and foreshortening. To achieve this, we made sure to appropriately choose the tomographic lines investigating the pixels where the radar image was pure, i.e., where the considered resolution cell was produced by the contribution of a single scattering effect, and not by the sum of several effects, due to the particular slant projection geometry and the plane wave approximation. In order to obtain the best metric interpretation of the tomographic experimental results, it is necessary to consider that the horizontal axis of each tomogram is that of the tomographic lines and follows the resolution of the radar, while the vertical axis is estimated through the sound focusing algorithm and therefore is applied by algorithm (24). In this context, it is possible to state without any reasonable doubt that each pixel corresponds to 1 m of spatial resolution in space.

This subsection describes all the tomographic measurements, and the entire internal architecture of Khnum-Khufu has been redesigned, which we propose in Figure 22. The complete list of rooms, corridors and tunnels that had never been inventoried until now is shown in Table 3. Each structure (simple or complex) was assigned to a unique tag numbered from 1 to 20, according to Figure 22. For this work, we have not used any kind of simulated data or predictive mathematical model, but rather, we report in a scientific manner what the CSG satellite has brought to our attention. Here, we list the explanation of all the results obtained respecting the order given in Table 3. Each structure listed in Table 3 will be described in detail, identified in the tomography and reconstructed within a Computer-Aided Design and Drafting (CAD) environment where all measurements will be provided. For the sake of clarity, we have included Table 4 that has the task of connecting each structure, tagged with a unique number increasing from 1 to 20, through its description, to the tomographic figure on which it is detected.

**Table 3.** List of discovered structures.

| Structure Number | Structure Type | Structure Name |
| --- | --- | --- |
| 1 | Corridor | Eastern ascending ramp |
| 2 | Corridor | Western ascending ramp |
| 3 | Corridor | Southern corridor |
| 4 | Corridor | Eastern descending ramp |
| 5 | Corridor | Western descending ramp |
| 6 | Corridor | Northern underground corridor |
| 7 | Corridor | Northern–East underground corridor |
| 8 | Corridor | Northern–West underground corridor |
| 9 | Complex structure | Northern underground complex structure |
| 10 | Complex structure | Zed complex structure |
| 11 | Room | Eastern sarcophagus passage facility |
| 12 | Room | Western sarcophagus passage facility |
| 13 | Room | Bottom sarcophagus room facility |
| 14 | Room | Queen's bottom room |
| 15 | Room | Southern bottom room |
| 16 | Corridor | Southern connection |
| 17 | Room | Little void |
| 18 | Corridor | Front corridor |
| 19 | Room | Big void |
| 20 | Complex structure | Zed–big void double connection |

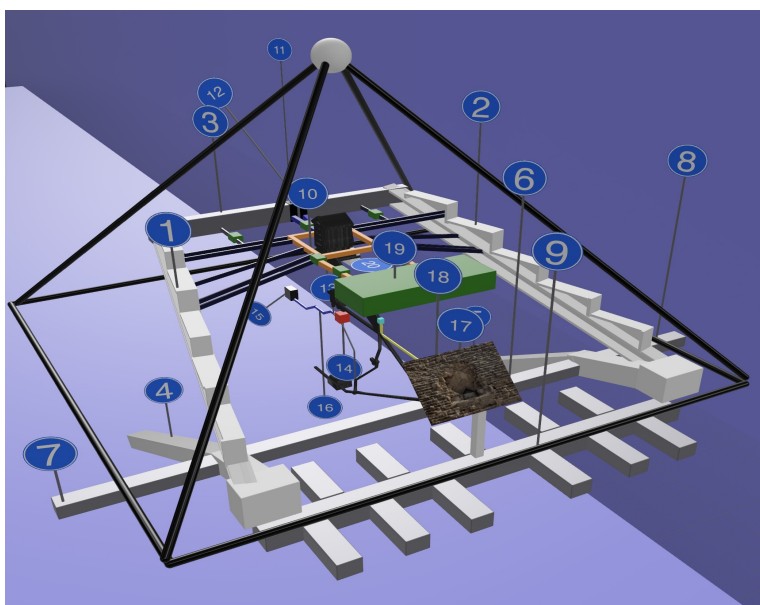

**Figure 22.** Three-dimensional (3D) reconstruction of the pyramid of Khnum-Khufu downstream of the interpretation of tomograms estimated through radar ultrasound.

### 5.3. Imaging of Known Structures

Before describing what has been discovered (what is still unknown), we propose the imaging of the known objects, in particular the King's room, the Zed, the Queen's room, the Grand Gallery, the grotto, and the so-called unfinished room. The SAR image of the pyramid is shown in Figure 23a, while the internal diagram of the pyramid, oriented toward the north (the northern direction goes from left to right) is shown in Figure 23b. The first tomographic result is presented in Figure 24a,b. This result is illustrated in Figure 23a by means of two yellow lines (identified by the number 1) that extend from the ground toward the pyramid apex. By estimating the vibrations, the pyramid is transparent due to their penetration characteristics within the solid rock, and its internal structures can be observed in Figure 24a,b. Figure 24a represents the sonic tomography partially overlapped with the picture in Figure 23b, while Figure 24b shows the non-overlapped tomography where three areas of interest are shown, and the details are studied below. Figure 25a–c is the detailed representation of the Zed, located inside box number 1 of Figure 24b, where Figure 25a is the schematic representation, while Figure 25b,c are the partially overlapped and not-overlapped tomography magnitude images, respectively. The Queen's chamber particular is depicted in Figure 26a–c; here, we use the same representation strategy where Figure 26a is the room scheme and Figure 26b,c are the partially overlapped and not-overlapped tomography, respectively. Figure 27a,b is the detailed image contained within the red box 3 of Figure 24b, and the void commonly referred to as 'the Grotto' is clearly visible. The lower chamber, the one commonly referred to as 'the unfinished room', is detected by radar, although with a weak signal, within the white box number 2. The Queen's room appears to be connected through a corridor connecting the grotto to the room below. This corridor, although it appears to be abruptly interrupted, is detected through a radar signature, which is marked by the white arrow number 3. It is assumed that this corridor follows the trajectory indicated by the red line, which is also pointed out by the white arrow number 3. Figure 2b,c show the detailed figure of the Zed, the colossal monument located at the heart of the pyramid (visible in Figure 24a). Figure 2b is its representation along the west–east direction, while Figure 2c shows its pattern along the south–north direction. Concluding this section, we propose in Figure 28a,b the partially and not-overlapped tomography of the Zed, respectively, where also the King's chamber is visible.

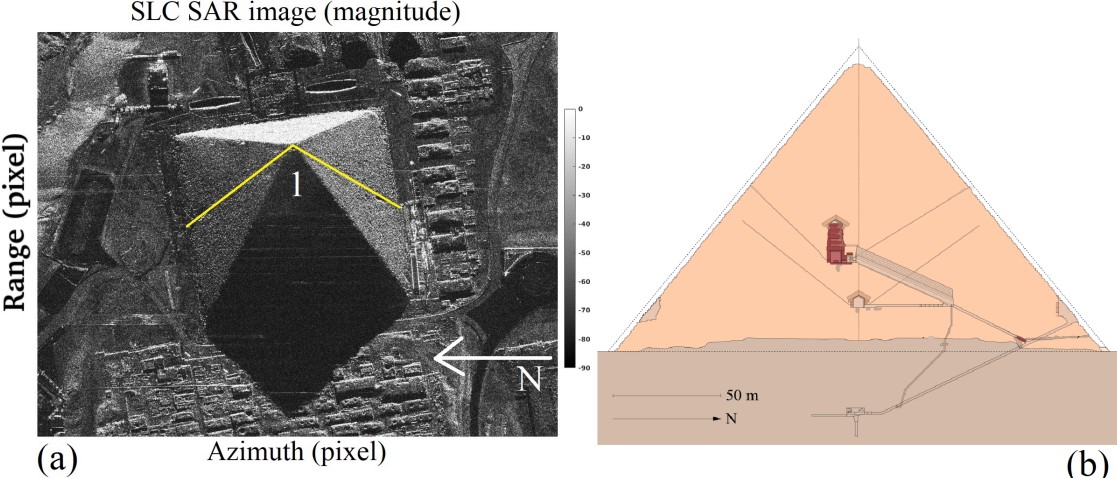

**Figure 23.** (**a**): SLC SAR of the pyramid of Khnum-Khufu. The V-shaped plot represents the tomographic line for which the vibrations were calculated, so that the inner section of the pyramid can be represented. (**b**): Internal scheme of the known structures.

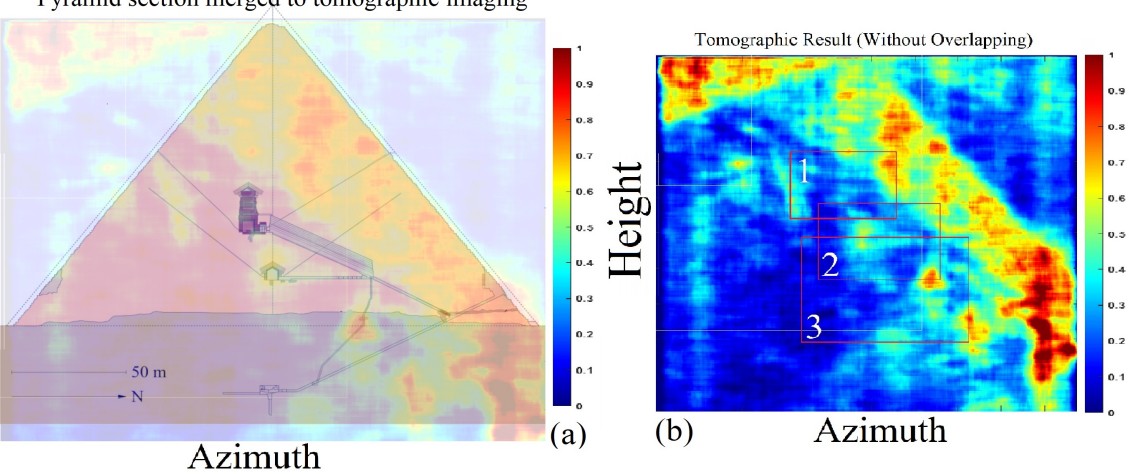

**Figure 24.** (**a**): Tomographic map of Khnum-Khufu overlapped to its schematic representation. (**b**): Tomographic map of Khnum-Khufu.

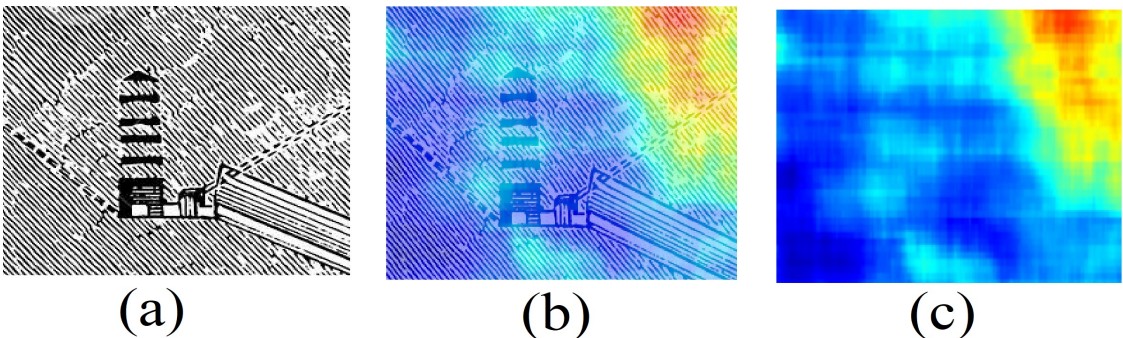

**Figure 25.** (**a**): Schematic representation of the Zed (box 1 details of Figure 24b). (**b**): Schematic representation of the Zed (box 1 details of Figure 24b) partially overlapped to tomographic result. (**c**): Figure 24b box 1 details tomographic result.

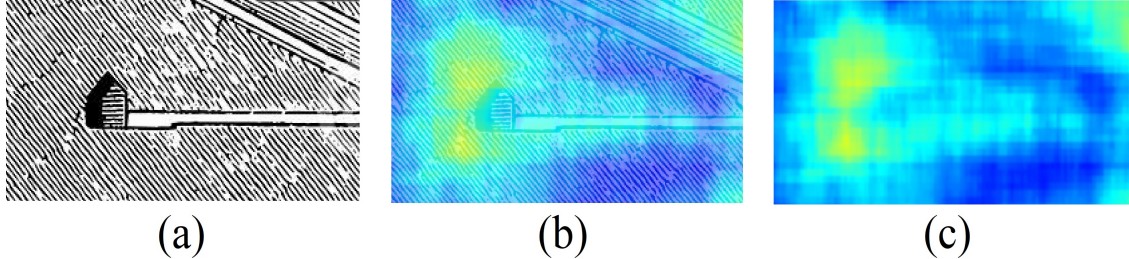

**Figure 26.** (**a**): Schematic representation of the queen's chamber (box 2 details of Figure 24b). (**b**): Schematic representation of the queen's chamber (box 2 details of Figure 24b) partially overlapped to tomographic result. (**c**): Figure 24b box 2 details tomographic result.

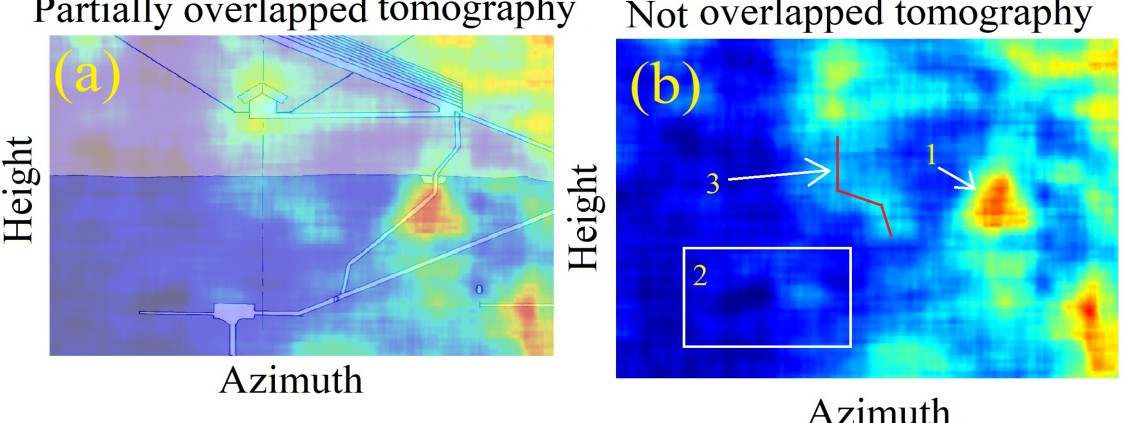

**Figure 27.** (**a**): Schematic representation of Figure 24b box 3 details partially overlapped with tomographic result. (**b**): tomographic result of Figure 24b box 3 details.

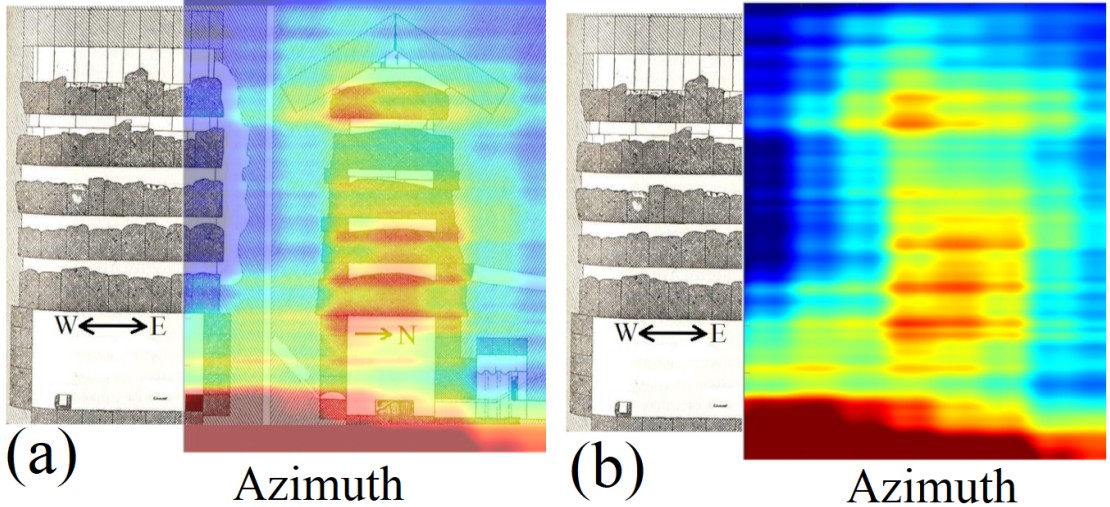

**Figure 28.** Schematic representation of the Zed and the King chamber. (**a**): Schematic representation of the Zed and the King chamber with partially-overlapped tomographic result. (**b**): Schematic representation of the Zed and the King chamber with totally-overlapped tomographic result.

Here, we show in sequence all the tomographies we have calculated, Figures 29, 30, 31, 32, 33, 34, 35, 36, 37 and 38a represents the SAR image in magnitude, whereby the tomographic line of investigation is visible (indicated with a yellow line marked with the number 1), and Figures 29, 30, 31, 32, 33, 34, 35, 36, 37 and 38b represents the tomograms (again in magnitude). In order to be more clear, we report the aforementioned figures in Table 5.

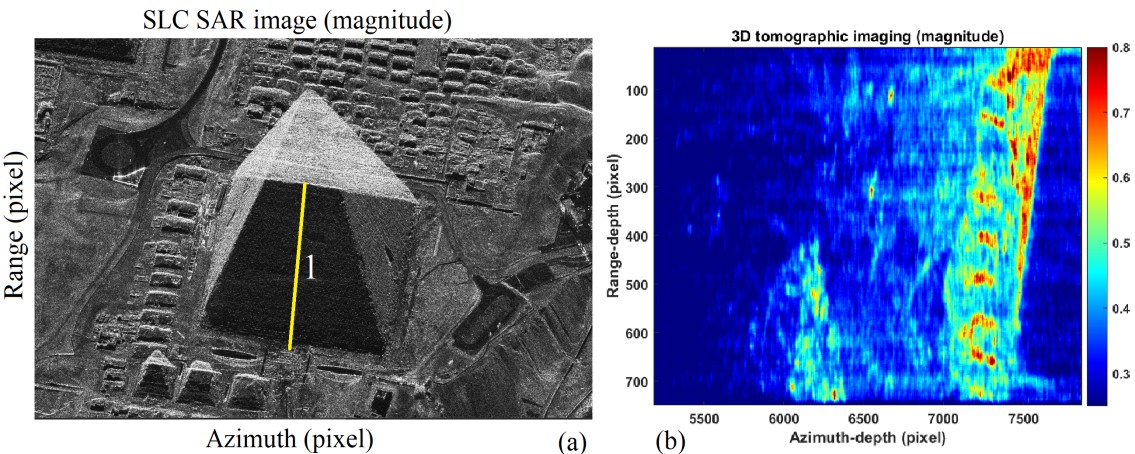

**Figure 29.** SAR images. (**a**): SLC SAR image (magnitude). The tomographic line 1 is oriented along the Eastern side of the pyramid. (**b**): Tomographic result (magnitude).

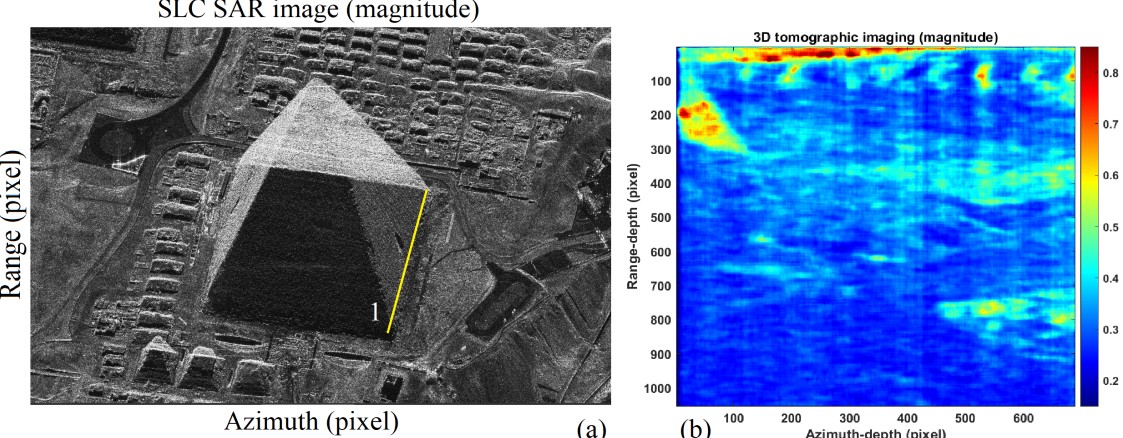

**Figure 30.** SAR images. (**a**): SLC SAR image (magnitude). The tomographic line 1 is oriented along the Northern-side of the pyramid. (**b**): Tomographic result (magnitude).

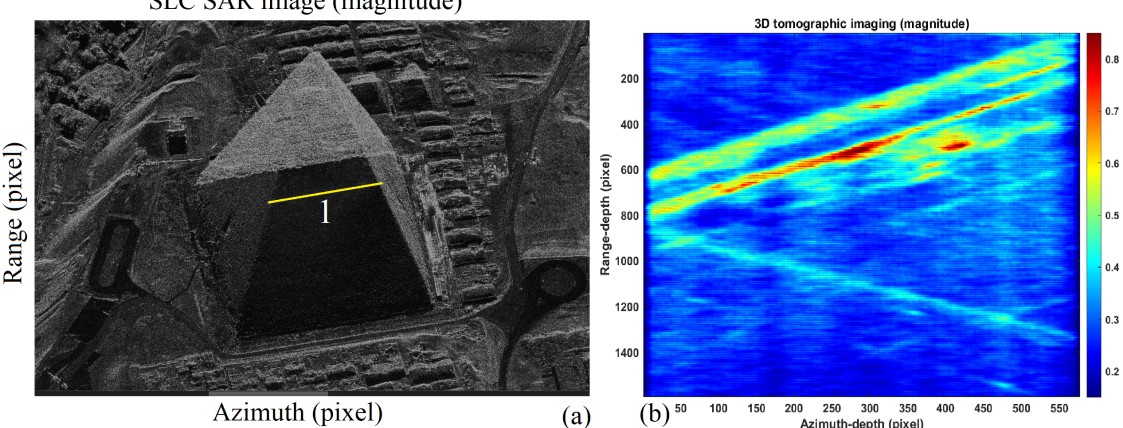

**Figure 31.** SAR images. (**a**): SLC SAR image (magnitude). The tomographic line 1 is oriented along the Western-side of the pyramid. (**b**): Tomographic result (magnitude).

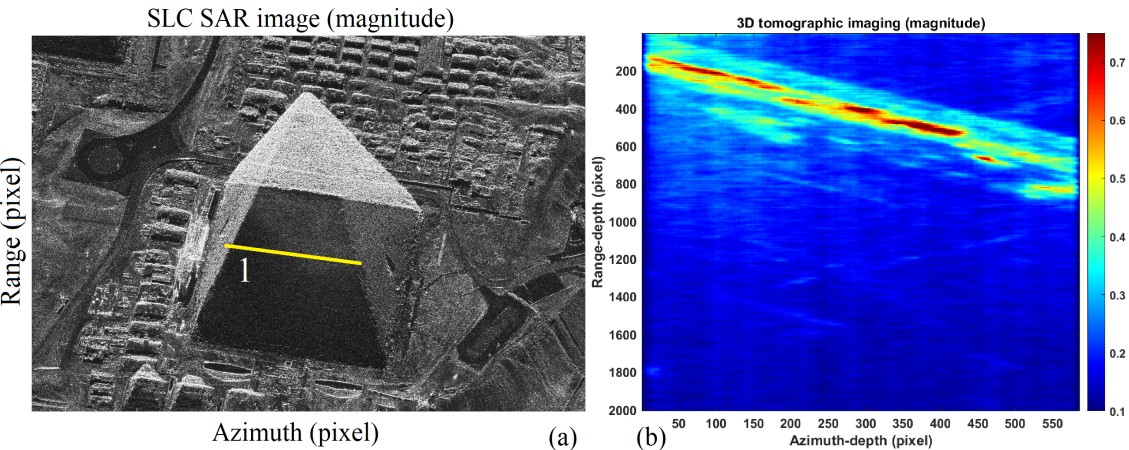

**Figure 32.** SAR images. (**a**): SLC SAR image (magnitude). The tomographic line 1 is oriented along the Eastern-side of the pyramid. (**b**): Tomographic result (magnitude).

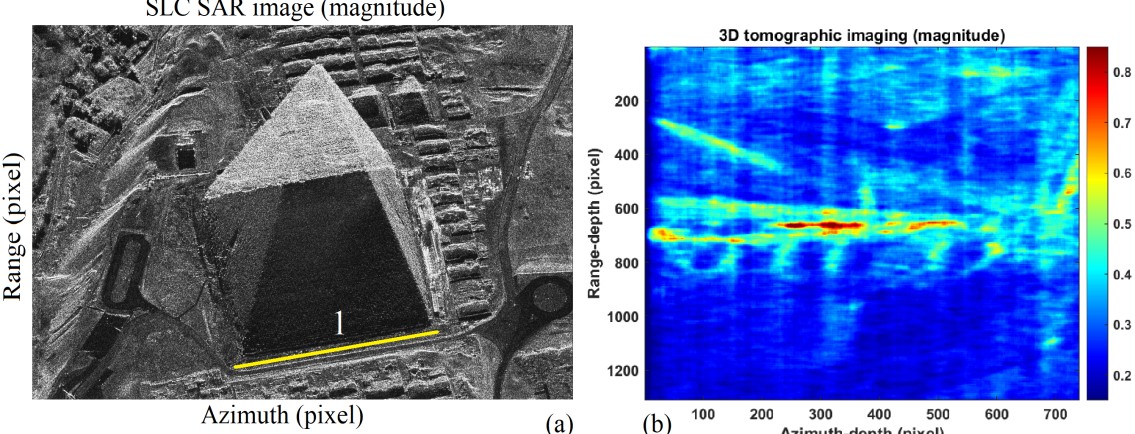

**Figure 33.** SAR images. (**a**): SLC SAR image (magnitude). The tomographic line 1 is oriented along the Western-side of the pyramid. (**b**): Tomographic result (magnitude).

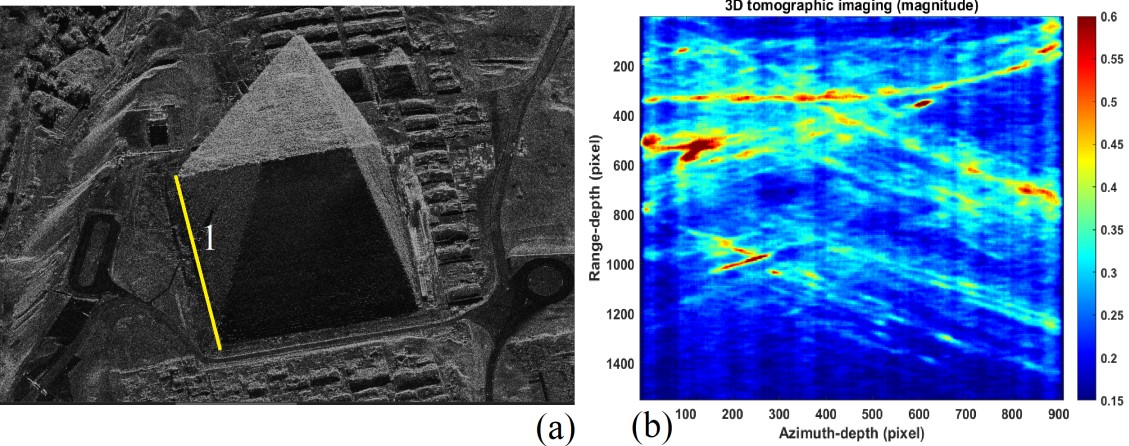

**Figure 34.** SAR images. (**a**): SLC SAR image (magnitude). The tomographic line 1 is oriented along the Northern-side of the pyramid. (**b**): Tomographic result (magnitude).

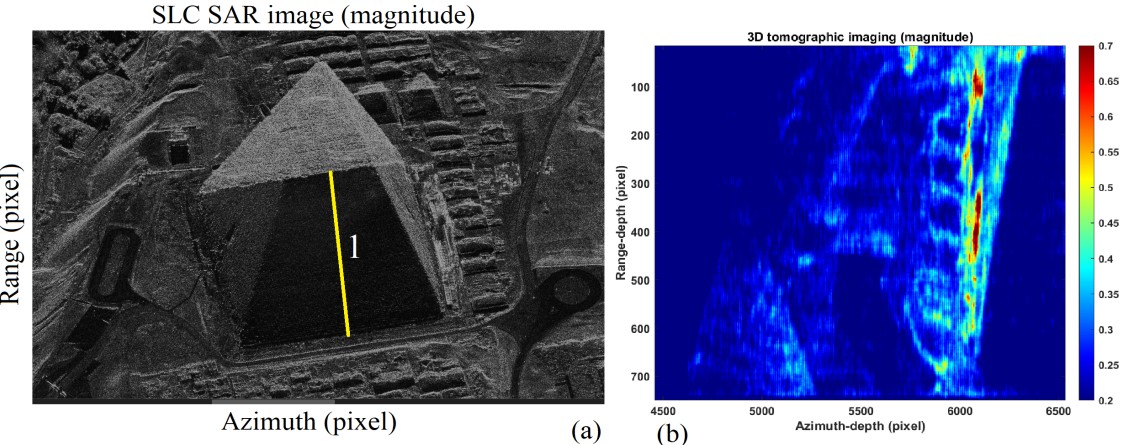

**Figure 35.** SAR images. (**a**): SLC SAR image (magnitude). The tomographic line 1 is oriented along the Western-side of the pyramid. (**b**): Tomographic result (magnitude).

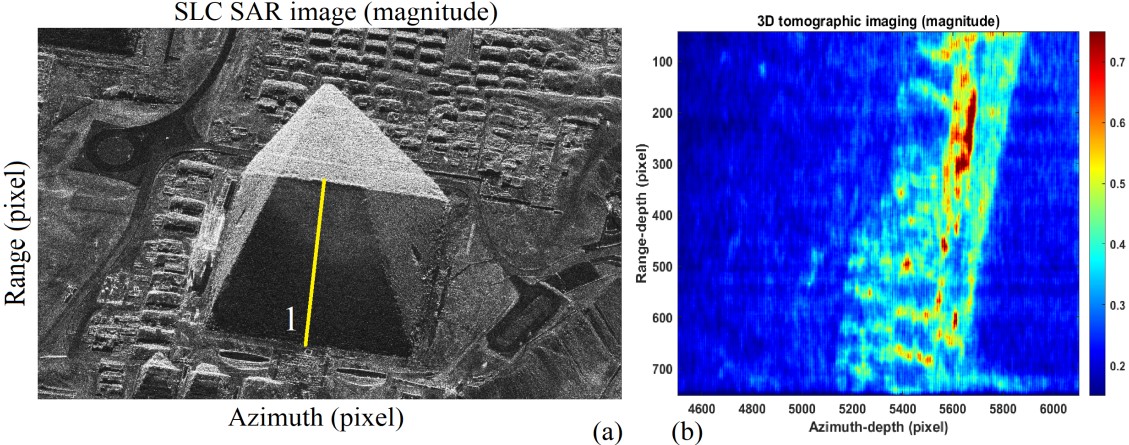

**Figure 36.** SAR images. (**a**): SLC SAR image (magnitude). The tomographic line 1 is oriented along the Eastern-side of the pyramid. (**b**): Tomographic result (magnitude).

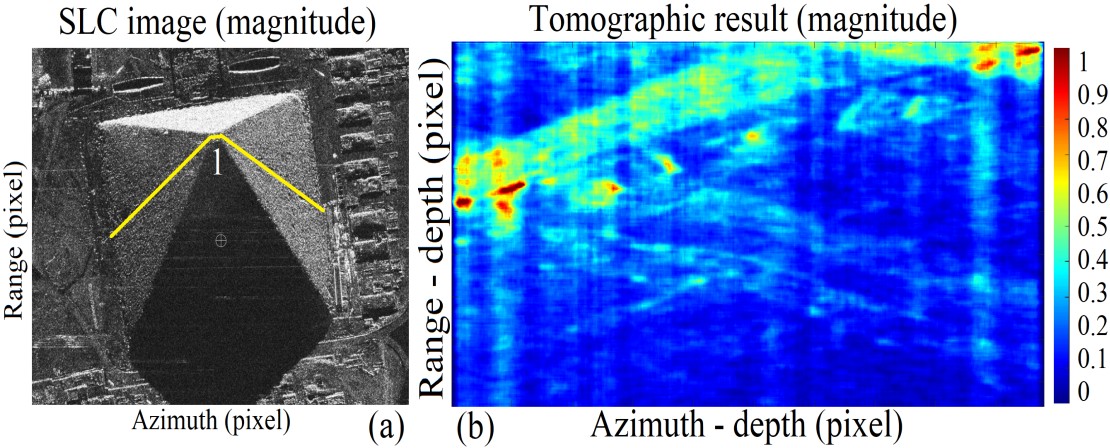

**Figure 37.** SAR images. (**a**): SLC SAR image (magnitude). The tomographic line 1 is oriented along the Northern-Southern-sides of the pyramid. (**b**): Tomographic result (magnitude).

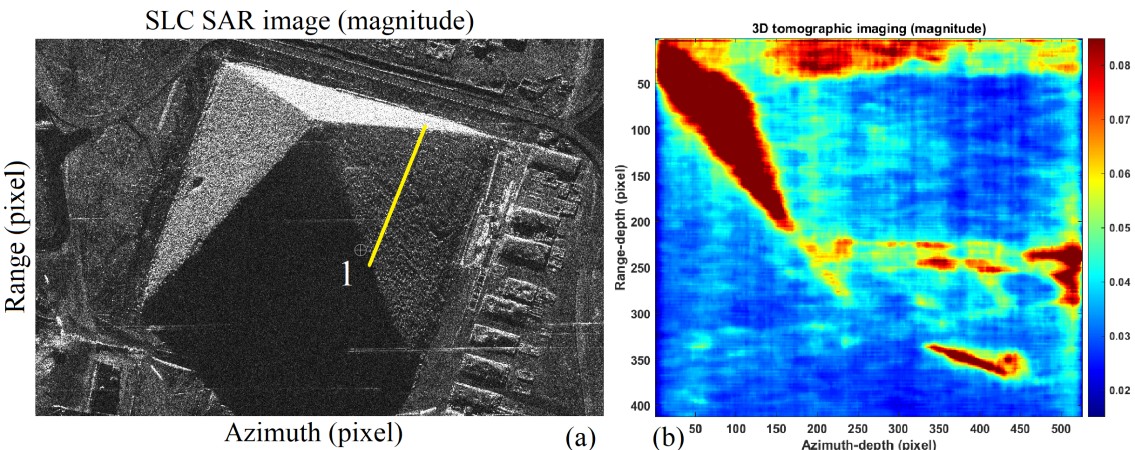

**Figure 38.** SAR images. (**a**): SLC SAR image (magnitude). The tomographic line 1 is oriented along the Southern-side of the pyramid. (**b**): Tomographic result (magnitude).

### 5.4. Eastern and Western Ascending Ramps (Tag 1, Tag 2)

Two inclined and diverging ramps (identified with the numbers 1 and 2 in the 3D reconstruction depicted in Figure 22), characterized by an approximate slope of about 42 degrees, are located inside the west and east sides. For both ramps, the lower part starts from the ground level on the north side and reaches half the height of the pyramid on the south side. The reference images are Figure 39a,b, for the eastern side and Figure 40a,b, for the western side, where the 3D models are compared to measured tomograms. From the figures, tags number 1, 2, 3, 4, 5, 7 and 9 are recognized.

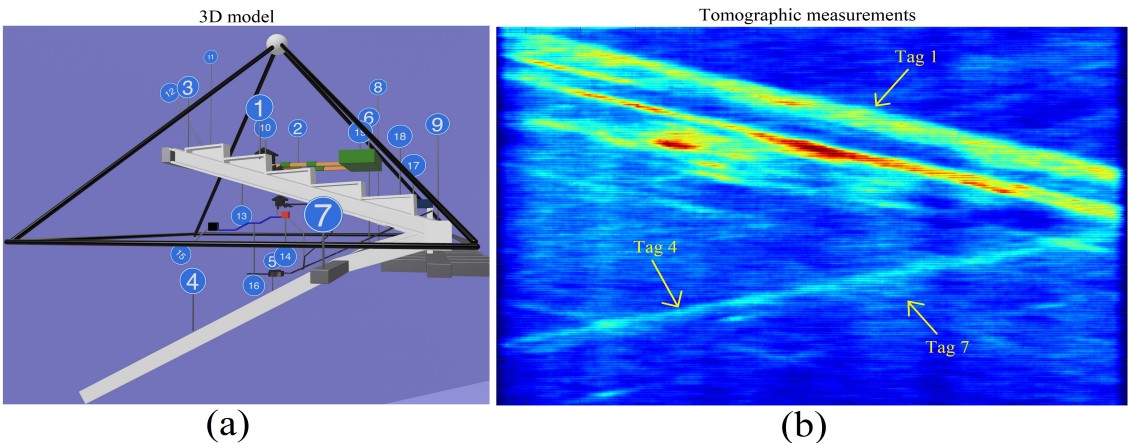

**Figure 39.** Tags association from tomography to 3D model. (**a**): Three-dimensional (3D) model of Khnum-Khufu (East-side orientation). (**b**): Tomographic reconstruction (magnitude).

### 5.5. Southern Corridor (Tag 3)

The ascending corridors are connected to each other by means of a horizontal structure placed at a height of about 90 m and located near the south side of the pyramid (identified with the number 3 in Figure 22). The corridor is recognized in Figure 41b, where the 3D reconstruction can be seen in Figure 41b.

### 5.6. Eastern and Western Descending Ramps (Tag 4, Tag 5)

Two ramps that are both connected to the previous ones run parallel to each other and also to the east and west base sides that run through a descending underground section with variable slope (numbers 4 and 5 in the 3D model). Figures 42a and 43a are the 3D reconstruction models showing the descending corridors from two different view angles,

while Figures 42b and 43b corresponds to tomographic measurements of the same tags, corresponding to the same descending corridors.

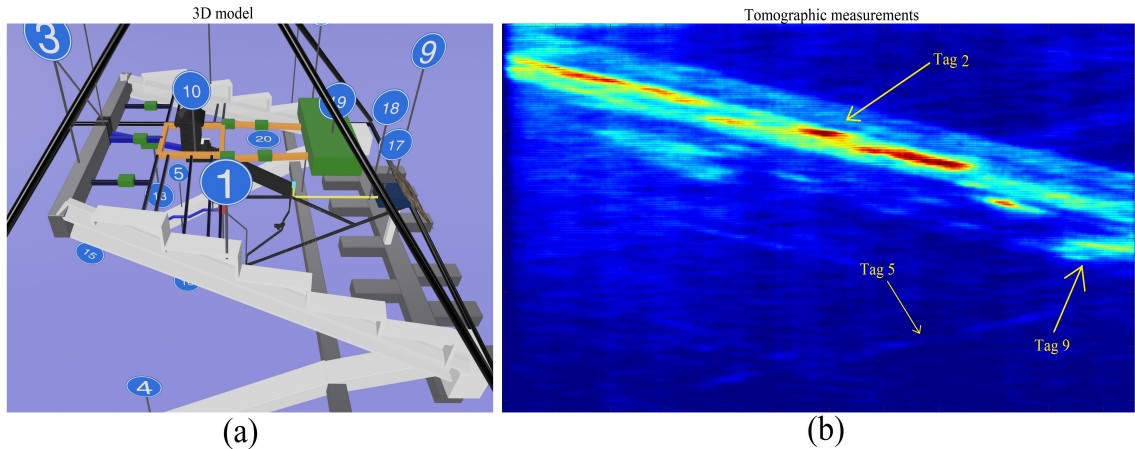

**Figure 40.** Tags association from tomography to 3D model. (**a**): Three-dimensional (3D) model of Khnum-Khufu (Top-East-side orientation). (**b**): Tomographic reconstruction (magnitude).

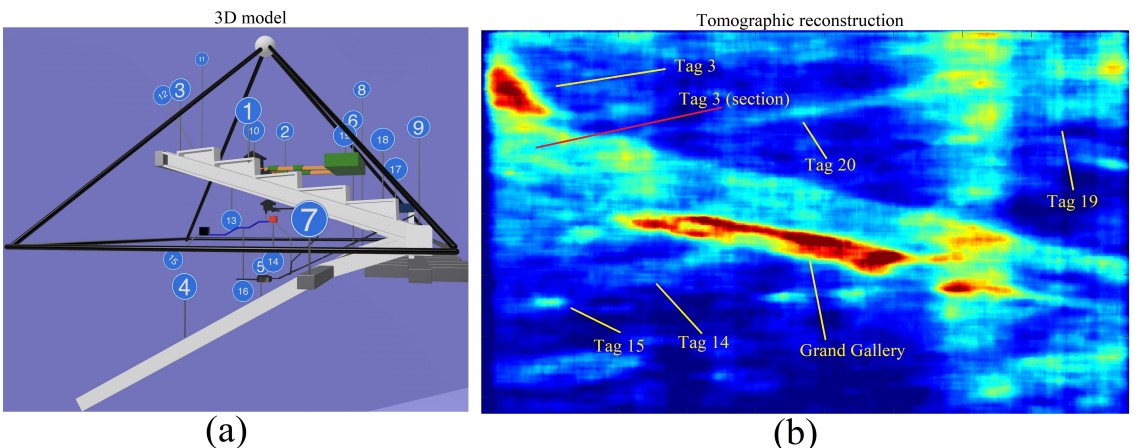

**Figure 41.** Tags association from tomography to 3D model. (**a**): Three-dimensional (3D) model of Khnum-Khufu (East-side orientation). (**b**): Tomographic reconstruction (magnitude).

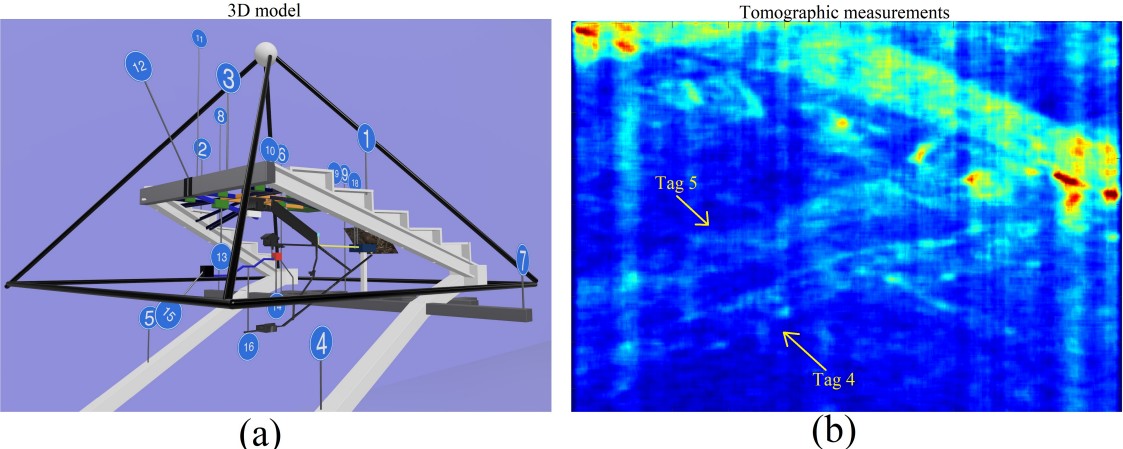

**Figure 42.** Tags association from tomography to 3D model. (**a**): Three-dimensional (3D) model of Khnum-Khufu (South-East-side orientation). (**b**): Tomographic reconstruction (magnitude).

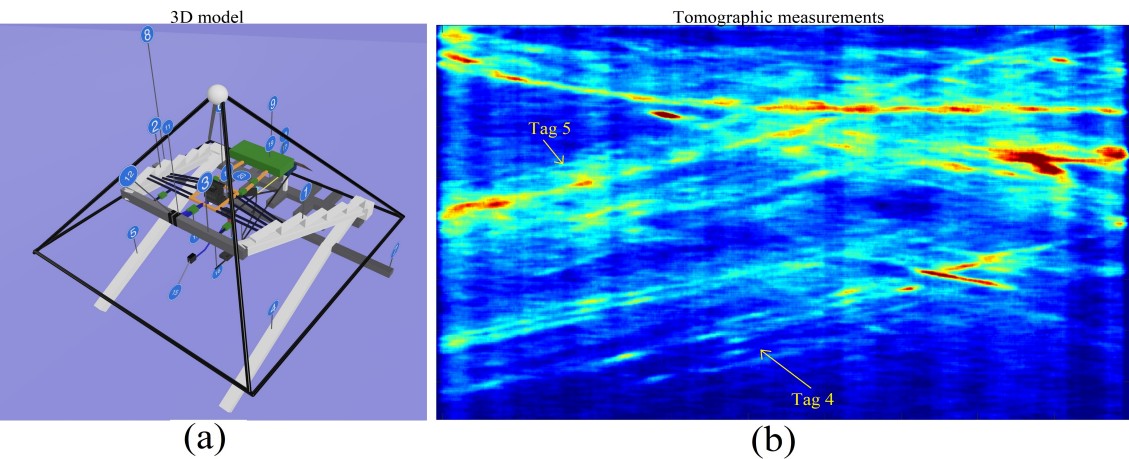

(a)                                                                    (b)

**Figure 43.** Tags association from tomography to 3D model. (**a**): Three-dimensional (3D) model of Khnum-Khufu (South-East-Top-side orientation). (**b**): Tomographic reconstruction (magnitude).

### 5.7. Northern Underground Corridor (Tag 6) and Northern–East and Northern–West Underground Corridors (Tag 7, Tag 8)

At the point where the descending ramps (tags 4 and 5) increase slope, they appear connected with the northern underground corridor (tag number 6 structure), parallel to the north side of the pyramid. The northern underground corridor is characterized by two extrusions, which are still located underground. Figure 44a is the 3D reconstruction model showing the structures tagged by numbers 6, 7 and 8, while Figure 44b corresponds to tomographic measurements of the same tags, corresponding to the same descending corridors. The section of this structure is deducted on the tomogram of Figure 45b, while the corresponding 3D model is showed in Figure 45a.

### 5.8. Northern Underground Complex Structure (Tag 9)

Immediately below the base of the pyramid structure, at the north side, a complex structure appears consisting of a horizontal body from which several identical bodies branch off, extruded perpendicularly to the main structure and characterized by a geometry, also present in other Egyptian pyramids, such as the pyramid of Zawyet El-Aryan [7,8,11] and the Sekhemkhet pyramid [10]. This complex structure (Number 9 in the 3D model) is characterized by a small conduit placed in a central position that runs a short distance in a vertical direction, in analogy with the presence of a similar building also in El-Aryan [7,8,11] and Saqqara [10]. The reference tomography is shown in Figure 46b, while the 3D model is depicted in Figure 46a.

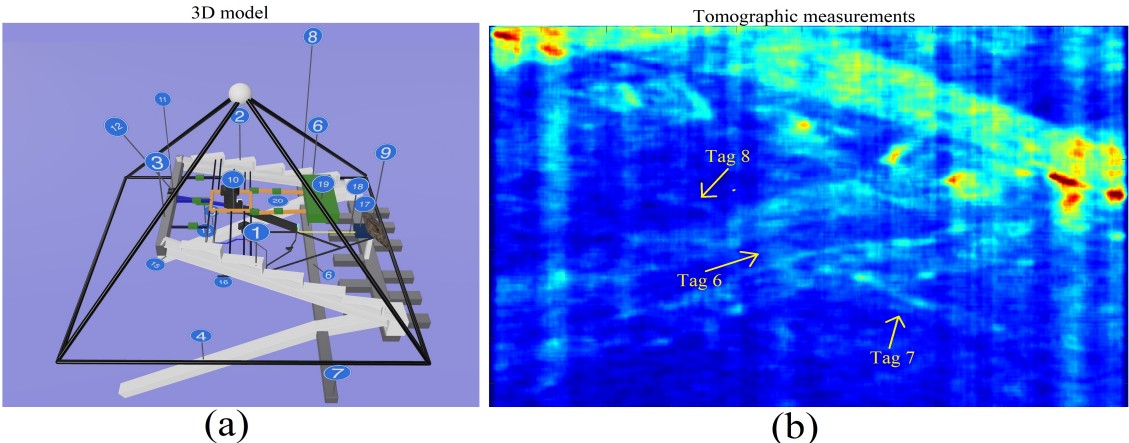

(a)                                                                    (b)

**Figure 44.** Tags association from tomography to 3D model. (**a**): Three-dimensional (3D) model of Khnum-Khufu (Top-East-side orientation). (**b**): Tomographic reconstruction (magnitude).

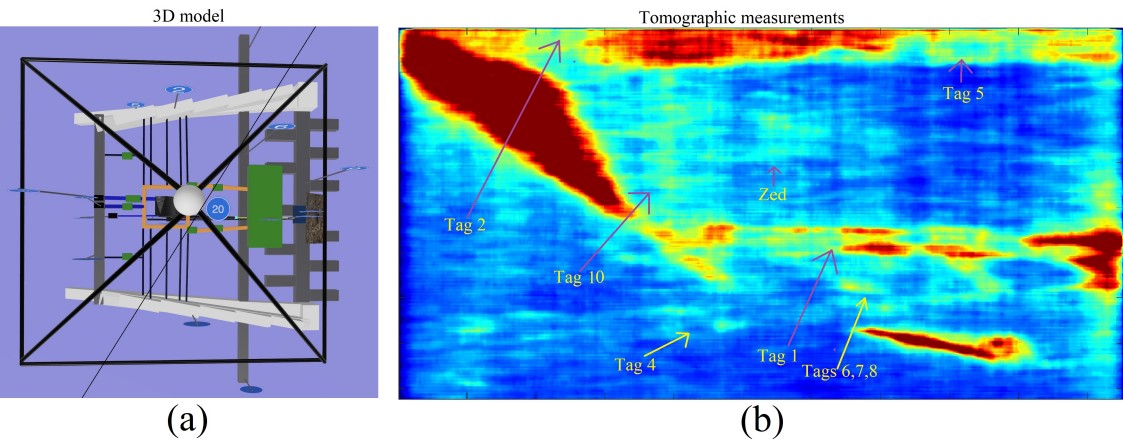

**Figure 45.** Tags association from tomography to 3D model. (**a**): Three-dimensional (3D) model of Khnum-Khufu (Top-side orientation). (**b**): Tomographic reconstruction (magnitude).

**Table 4.** List of discovered structures detected in figures.

| Structure Number | Structure Name | Detected in Figures |
|---|---|---|
| 1 | Eastern ascending ramp | Figures 39, 45, 47 and 56. |
| 2 | Western ascending ramp | Figures 40, 45, 47 and 56. |
| 3 | Southern corridor | Figures 41 and 54–56. |
| 4 | Eastern descending ramp | Figures 41–43, 45, 47 and 56. |
| 5 | Western descending ramp | Figures 40, 42, 43, 45 and 47. |
| 6 | Northern underground corridor | Figures 44 and 45. |
| 7 | Northern–East underground corridor | Figures 39, 44 and 45. |
| 8 | Northern–West underground corridor | Figures 44 and 45. |
| 9 | Northern underground complex-structure | Figures 40 and 46. |
| 10 | Zed complex structure | Figures 45, 48, 55 and 56. |
| 11 | Eastern sarcophagus passage facility | Figure 49. |
| 12 | Western sarcophagus passage facility | Figure 49. |
| 13 | Bottom sarcophagus room facility | Figure 50. |
| 14 | Queen's bottom room | Figures 41 and 52. |
| 15 | Southern bottom room | Figures 41 and 52. |
| 16 | Southern connection | Figure 52. |
| 17 | Little void | Figures 51 and 54. |
| 18 | Front corridor | Figure 51. |
| 19 | Big void | Figures 41, 47, 55 and 56. |
| 20 | Zed–big void double connection | Figures 41 and 47. |

*5.9. Zed Complex Structure (Tag 10)*

A complex square structure (identified with the number 10 tag), which connects itself to structure number 11, belongs to the structure of passage number 3. The structure 10 develops around the Zed, approximately at the height of the lowest room (Davison's Chamber) [17]. The reference tomography is shown in Figure 45b, while the 3D model is depicted in Figure 45a. The structure is also detected through different tomograms depicted in Figure 48b,c, where the reference SLC image is shown in Figure 48a, and the tomographic lines 1 and 2 are shown on the northern and southern pyramid surfaces, respectively.

*5.10. Eastern and Western Sarcophagus Passages Facility (Tag 11 and Tag 12)*

Structure number 3 also seems to contain two sub-structures, which are identified with the numbers 11 and 12, connected, through corridor 13, to the King's room, through a passage that seems located under the floor of the latter. The reference tomography is shown in Figure 49b, while the 3D model is depicted in Figure 49a.

*5.11. Bottom Sarcophagus Room Facility (Tag 13)*

A room located below structures 11 and 12, connecting facilities 3 to 13. The reference tomography is shown in Figure 50b, while the 3D model is depicted in Figure 50a.

The reference tomography is shown in Figure 52b, while the 3D model is depicted in Figure 52a.

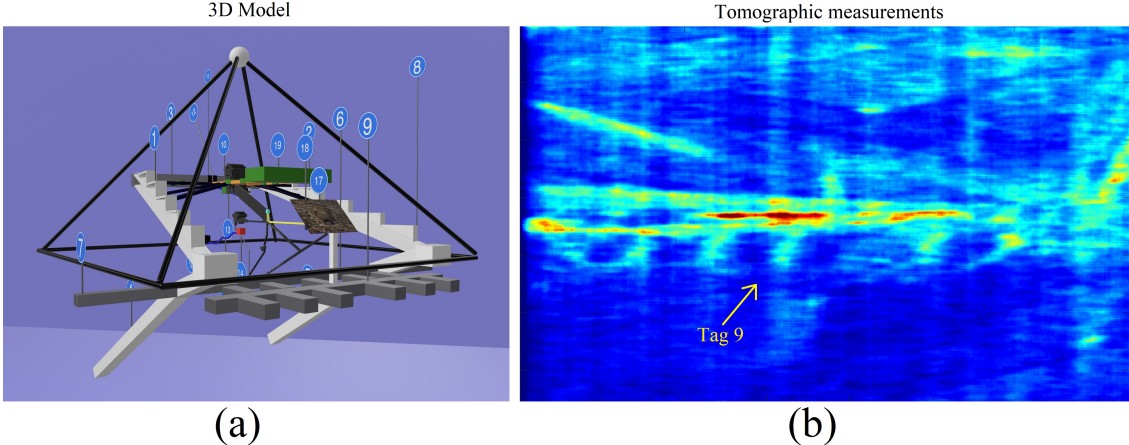

**Figure 46.** Tags association from tomography to 3D model. (**a**): Three-dimensional (3D) model of Khnum-Khufu. (**b**): Tomographic reconstruction (magnitude).

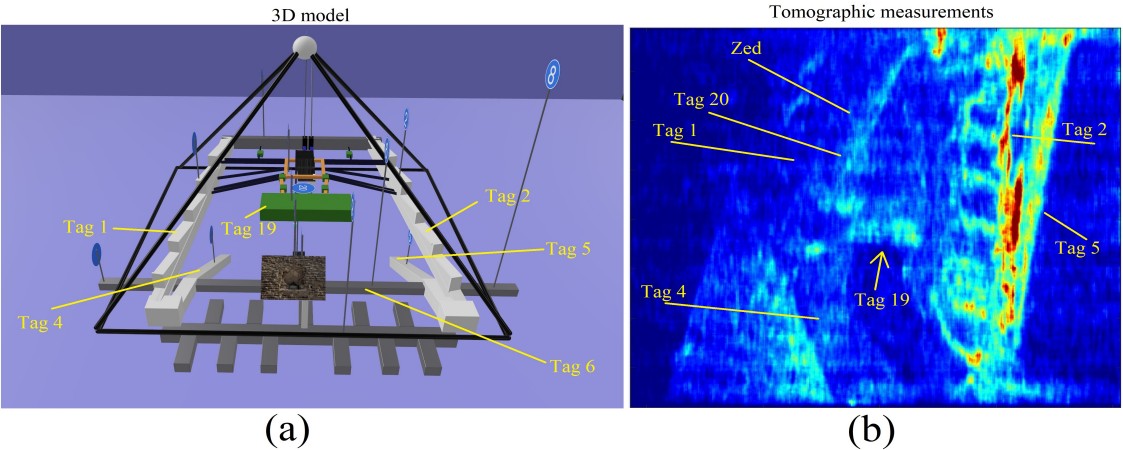

**Figure 47.** Tags association from tomography to 3D model. (**a**): Three-dimensional (3D) model of Khnum-Khufu. (**b**): Tomographic reconstruction (magnitude).

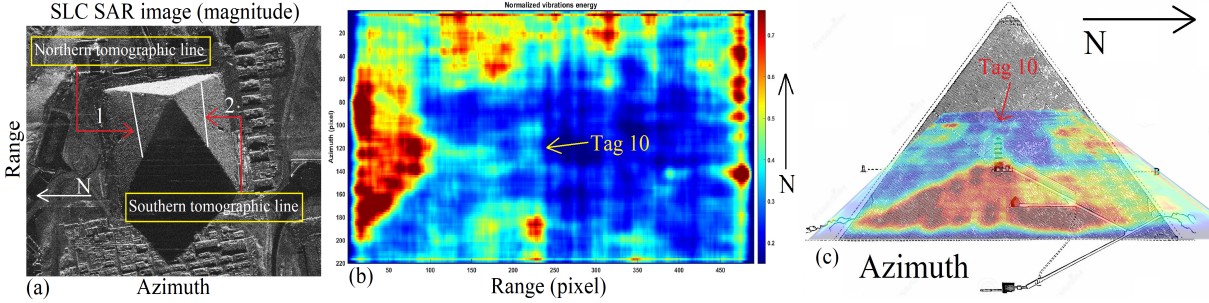

**Figure 48.** SLC SAR image (magnitude). (**a**): SAR image in magnitude. (**b**): Tomographic result (Top-side view). (**c**): Tomographic result (East-side view).

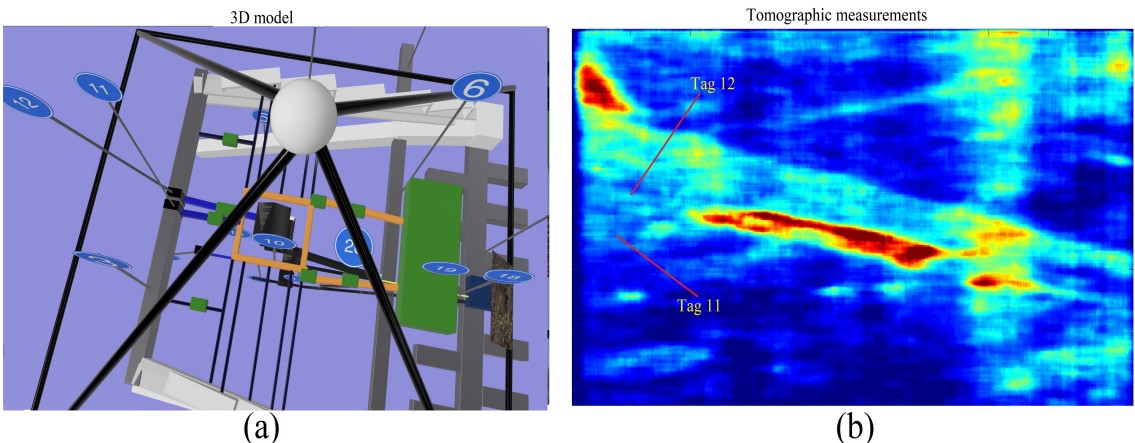

**Figure 49.** Tags association from tomography to 3D model. (**a**): Three-dimensional (3D) model of Khnum-Khufu. (**b**): Tomographic reconstruction (magnitude).

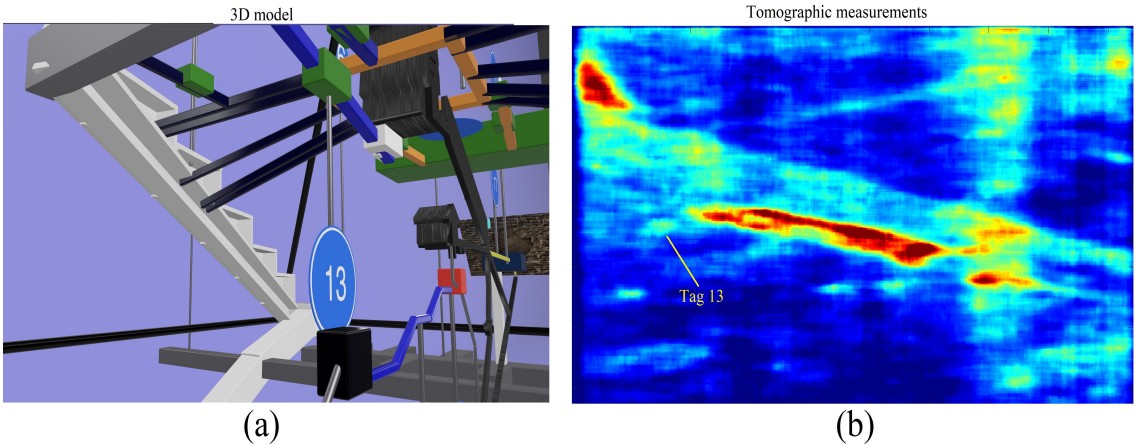

**Figure 50.** Tags association from tomography to 3D model. (**a**): Three-dimensional (3D) model of Khnum-Khufu. (**b**): Tomographic reconstruction (magnitude).

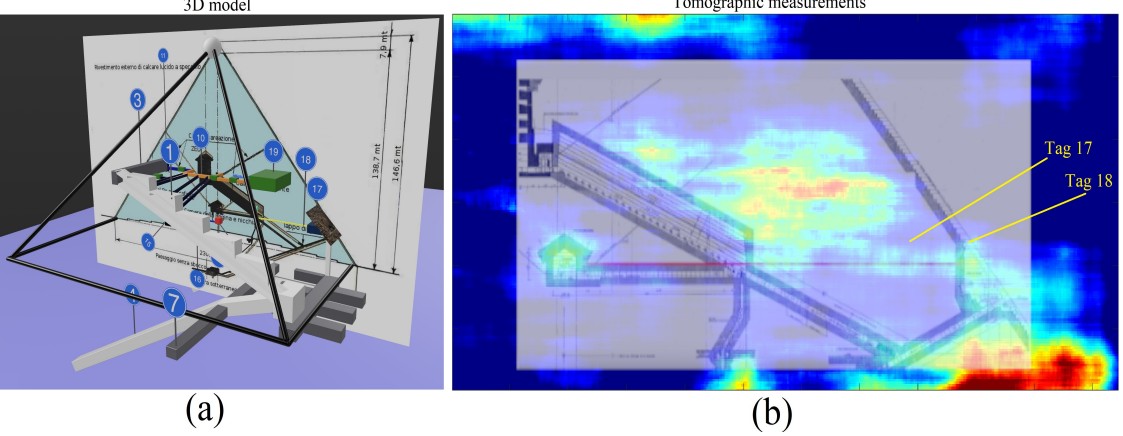

**Figure 51.** Tags association from tomography to 3D model. (**a**): Three-dimensional (3D) model of Khnum-Khufu. (**b**): Tomographic reconstruction (magnitude).

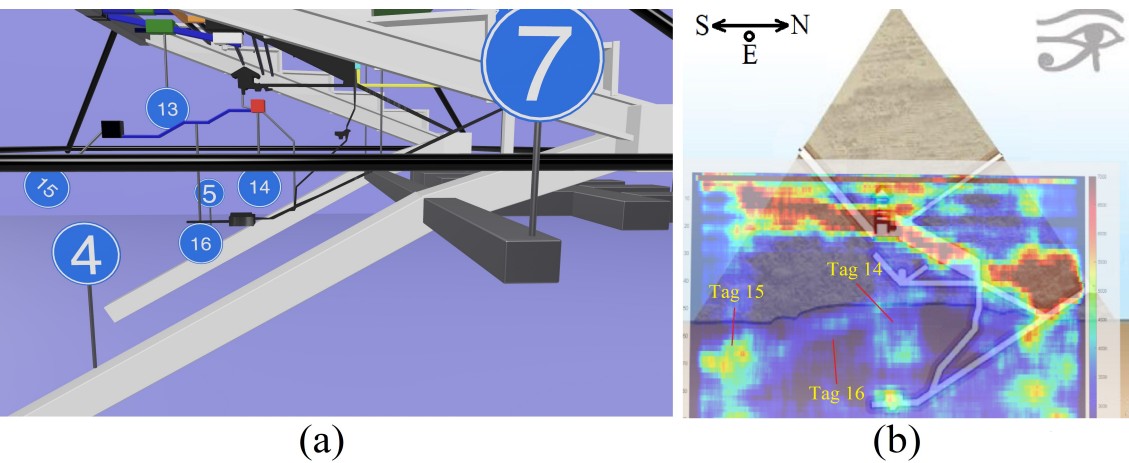

**Figure 52.** Tags association from tomography to 3D model. (**a**): Three-dimensional (3D) model of Khnum-Khufu. (**b**): Tomographic reconstruction (magnitude).

### 5.12. Queen's Bottom Room (Tag 14)

A further structure (identified with the number 14 in the 3D model) is located below the Queen's chamber and connected to it through a small conduit. From space number 14, the conduit seems to continue, bonding a similar path which, from the room known as the "Grotto", leads to the underground room called' 'unfinished. The reference tomography is shown in Figure 52b, while the 3D model is depicted in Figure 52a.

### 5.13. Southern Bottom Room (Tag 15)

A room is located on the bottom of structures 11 and 12. The reference tomography is shown in Figure 52b, while the 3D model is depicted in Figure 52a.

### 5.14. Southern Connection (Tag 16)

A further conduit (number 16 of the 3D model) joins the structure 14 to a structure, which is placed almost at ground level (number 15 of the 3D model). This facility has been discovered through the tomographic result depicted in Figure 52b, where a particular of the 3D model is visible in Figure 52a.

### 5.15. Little Void (Tag 17)

A void can be located immediately behind the original entrance of the pyramid, which is not easily identifiable in shape and size (number 17 of the 3D model), from which a horizontal conduit (number 18 of the 3D model) starts and which seems to end at the foot of the Grand Gallery but is not directly connected to it. This is a void located in front of the northern entrance of the pyramid [41]. The room is clearly visible in Figure 53b, which is precisely located above the corridor identified by the structure with tag 18. It possible to observe the little-void also on Figure 54b; there, the reference 2D model is reported in Figure 54a.

### 5.16. Front Corridor (Tag 18)

A corridor-like structure clearly visible in Figure 53b is located just behind the external V-shaped structure depicted in Figure 53a. The corridor is detected, and its tomographic representation is depicted in Figure 53b.

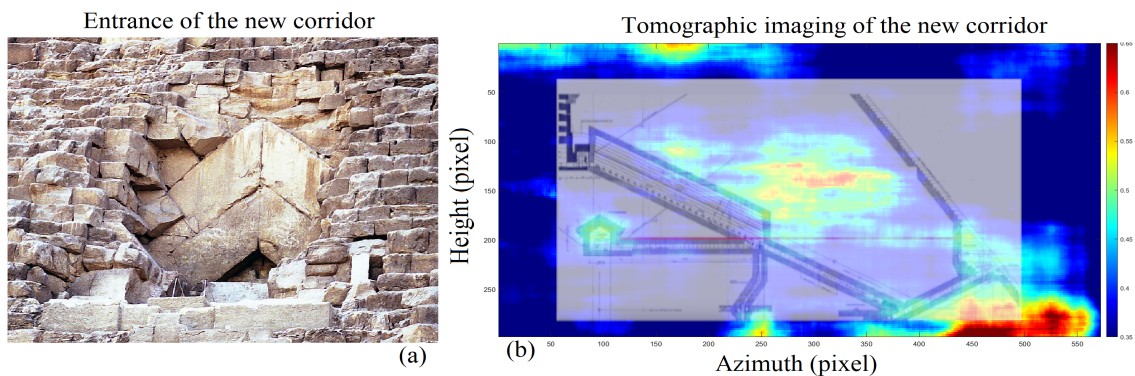

**Figure 53.** Entrance on Khnum-Khufu. (**a**): Optical image. (**b**): Tomographic map (magnitude).

*5.17. Big Void (Tag 19)*

This is a large structure whose shape resembles a parallelepiped (number 19 of the 3D model). This object appears to be connected to structure 10 by means of a double horizontal connection (number 20 of the 3D model). The reference tomography images are shown in Figures 55b and 56b, while the 3D models are depicted in Figure 55a and 56a. The large red target 1 visible in Figure 56b is a false alarm, which is generated by the southwest ascending angle of the pyramid [69].

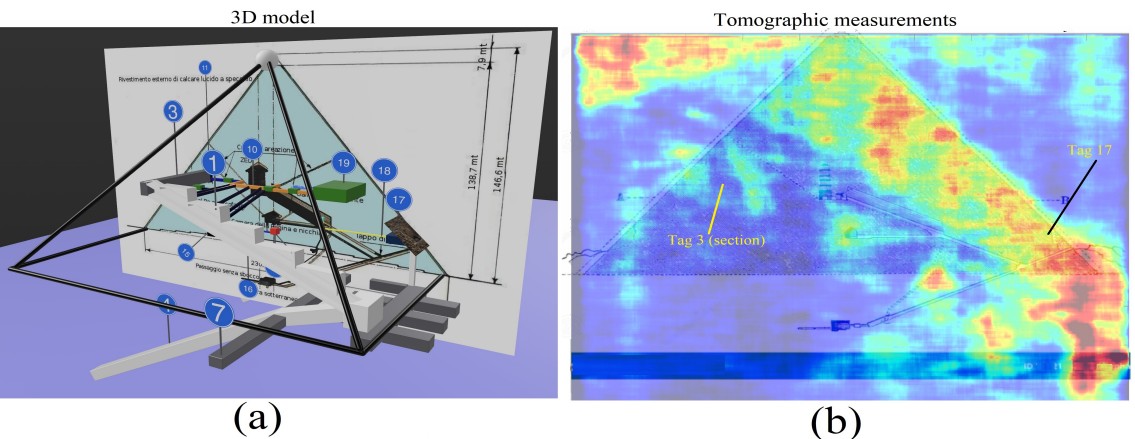

**Figure 54.** Tags association from tomography to 3D model. (**a**): Three-dimensional (3D) model of Khnum-Khufu. (**b**): Tomographic reconstruction (magnitude).

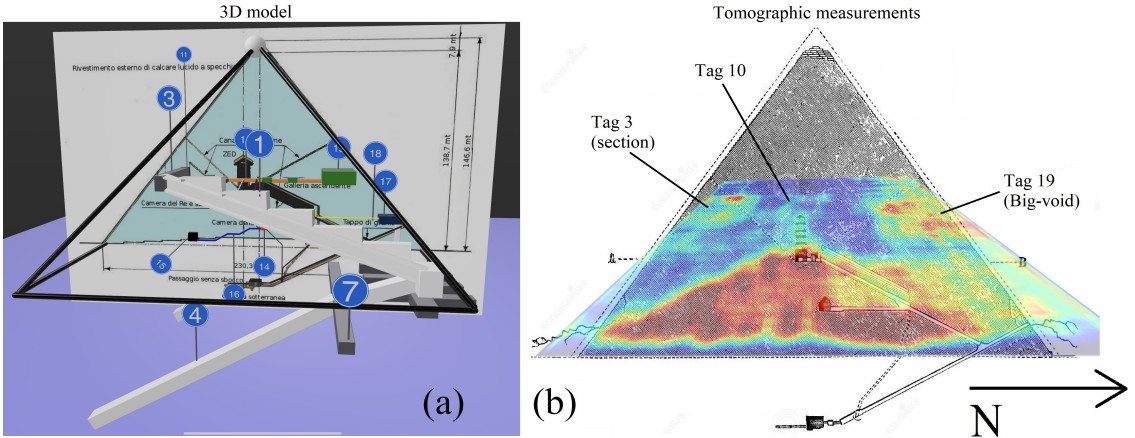

**Figure 55.** Tags association from tomography to 3D model. (**a**): Three-dimensional (3D) model of Khnum-Khufu. (**b**): Tomographic reconstruction (magnitude).

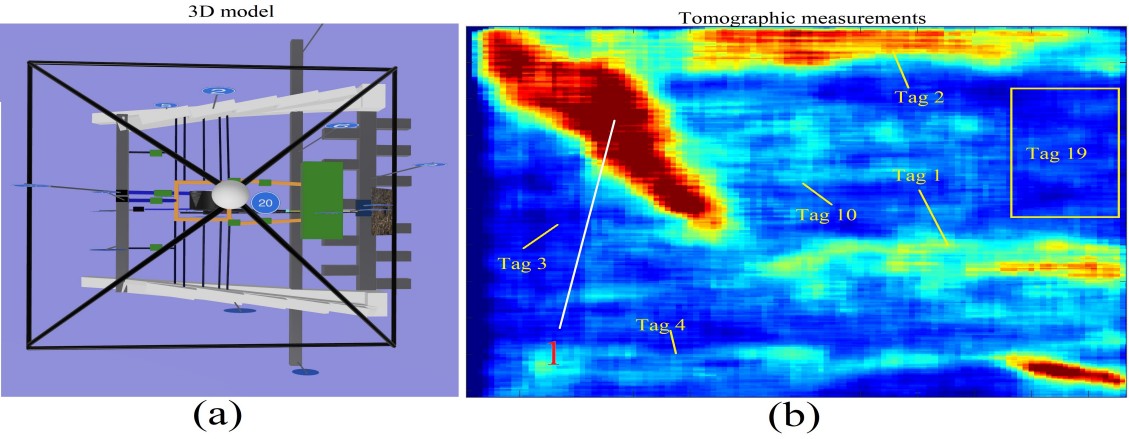

**Figure 56.** Tags association from tomography to 3D model. (**a**): Three-dimensional (3D) model of Khnum-Khufu. (**b**): Tomographic reconstruction (magnitude).

### 5.18. Zed–Big Void Double Connection (Tag 20)

Structure 19 (big void) is connected to the large corridor 3 to the south, at the height of the big void, via two oblique corridors. The reference tomography is shown in Figure 41b, while the reference 3D model is shown in Figure 49a.

### 5.19. Metric Determination

The final objective of this study is to provide approximate measurements of the structures detected using the Doppler SAR tomography technique. The measurements that we propose are expressed in meters and are affected by an error that we have estimated to be very low, with respect to the actual measurement of the structures, according to the particular methodology we used. The dimensions are proposed in Figures 57–59. The measurements we suggest also include the thickness of the material used to construct them and are not to be intended as mere empty space. With regard to obtaining only empty space, research is currently underway in order to improve the technique and find a way to distinguish solid spaces from hollow spaces. The measures reported are evaluated by using Tape Measuring Wall Area software (http://www.pictureenginecompany.com/MeasureEngine/Promo.html (accessed on 1 July 2022)) employing as internal standard the pyramid's base length and are in accordance with the results afforded by SAR data.

**Table 5.** List of the principal tomographic images.

| Picture | Tomographic Looking-Direction | Tomographic Line Orientation |
|---------|-------------------------------|------------------------------|
| Figure 29 | Eastern-side | Vertical |
| Figure 30 | Northern side | Horizontal |
| Figure 31 | Western side | Horizontal |
| Figure 32 | Eastern side | Horizontal |
| Figure 33 | Western side | Horizontal |
| Figure 34 | Northern side | Horizontal |
| Figure 35 | Western side | Vertical |
| Figure 36 | Eastern side | Vertical investigation |
| Figure 37 | Northern–Southern side | Vertical |
| Figure 38 | Southern side | Horizontal |

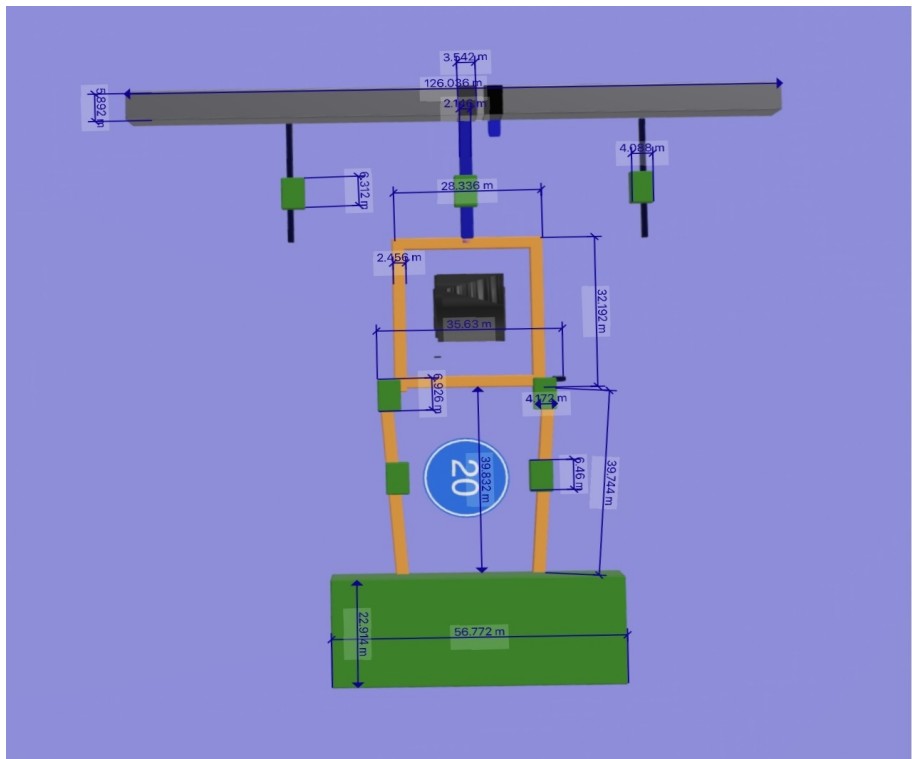

**Figure 57.** Measurements of the detected facilities of the pyramid. The numbers shown after the comma cannot be significant. (Top-side view).

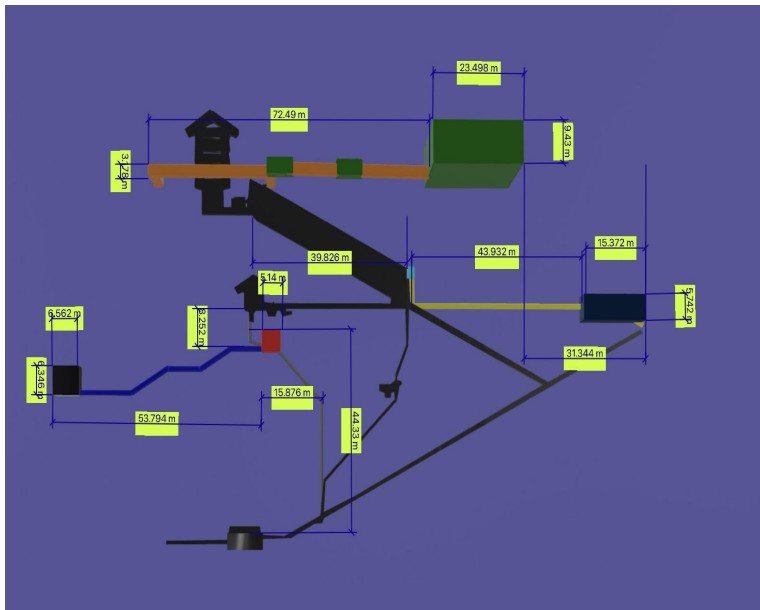

**Figure 58.** Measurements of the detected facilities of the pyramid. The numbers shown after the comma cannot be significant. (East-side view).

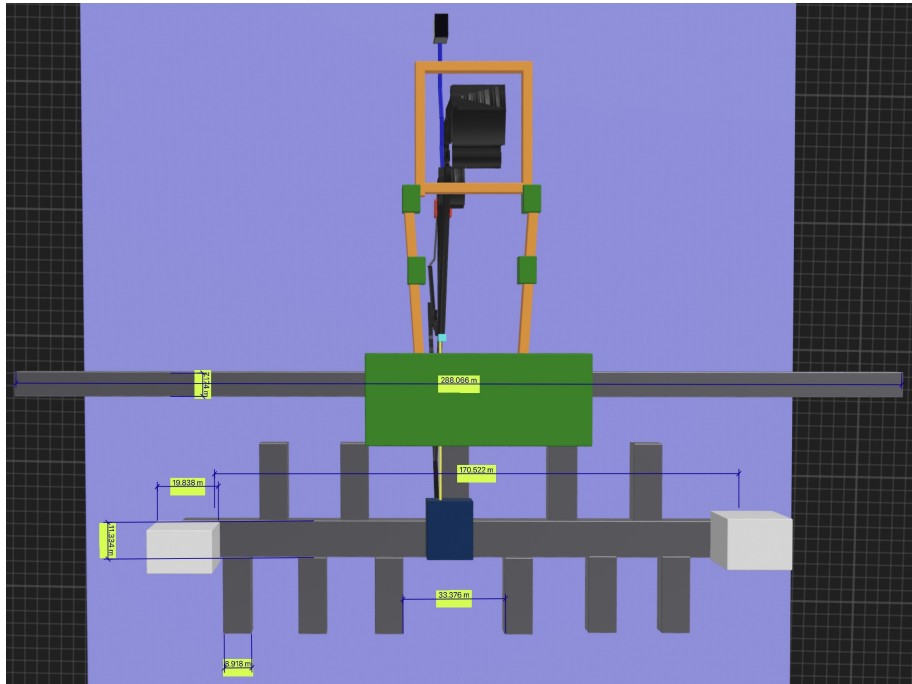

**Figure 59.** Measurements of the detected facilities of the pyramid. The numbers shown after the comma cannot be significant. (Top-side particular view).

## 6. Discussion

### 6.1. Data Analysis

Here, we proceed to data analysis, while being aware that some aspects of the internal structure of the pyramid of Khnum-Khufu still need to be clarified; it is possible, in our opinion, to attribute a meaning to the internal structures of the monument, taking into account all the data relating to previous research in this field. It is possible to underline how our 3D reconstruction, even if for now it does not claim to indicate the true measurements of the objects shown (even if the SAR technique can accurately evaluate this parameter), is following some facts exposed in previous research. Some researchers have shown how on the north edge of the east side of the pyramid at ground level, there is a thermal anomaly that suggests the presence of a room and a corridor located a few meters from the external wall of the monument [36,38,39,70]. These data are in agreement with our analyses which predict, at that point, the presence of a room as a link between the two ramps 1 and 4 of our 3D model. The microgravimetry data, carried out on the pyramid by different research groups [33,34,36,37], can show us how, under the floor of the King's room, there is a lack of homogeneity possibly attributable to structures such as those hypothesized by us (see points 12 and 13 of the 3D model). The presence of rooms located under the King's room is also amply documented by many photographic findings on the Web. The presence of ramps, placed inside the pyramid and which we highlight with certainty for the first time (numbers 1, 2, 3, and 4 of the 3D model), had already been postulated [15] and partly detected by electrogravitic measurements carried out in 1998 [35]. The presence of further rooms located near the Queen's room had already been postulated in the past by some researchers [35,71]; however, this is based on intuitions not confirmed by objective evidence. The authors of [71] also postulated the presence of a horizontal passage placed between the original entrance of the pyramid and the Great Gallery, in place of structure number 17, which we highlight in this work; moreover, in this work, the presence of a room located immediately after the entrance to the great pyramid is highlighted. This void also seemed to be confirmed by muon spectroscopy [9] carried out by researchers at the University of Nagoya in 2017 and indicated with the name "small void". It must be strongly emphasized that while the analyses used up to now in the attempt to describe the objects inside the pyramid gave only the possibility of making indirect hypotheses, the SAR methodology

visibly produces direct evidence of the geometries inside the objects that can be analyzed. In contrast, the muon spectroscopy used in 2017 [9] would not have been deemed reliable by the Egyptian Supreme Council of Antiquities, providing controversial data [41]. No trace of the presence of a structure identified as a big void, put in evidence by muonic spectroscopy [9], was in our hands detected by SAR. On the other hand, the SAR technique allows us to make more observations using different starting geometries, thus being able to see the same structures inside the pyramid, from different points of view, and the only possibility for a mistake lies in the visual interpretation of the data obtained but does not invalidate the presence or absence of specific structural elements. The geometries of the objects highlighted by the SAR in this paper also make us reasonably exclude any errors due to the possible non-homogeneity of the materials employed during the construction of the pyramid interiors.

### 6.2. Data Interpretation

On the basis of the above, it appears necessary to provide a plausible explanation as a key to understanding the function of the structures found inside the pyramid, taking into account that this interpretation is only intended to be a starting point for further interpretative ideas that could arise from a serene discussion at the level of the scientific community. The authors' vision starts from what we have already previously published in [72] and widely discussed [12,16,22,23,28,31,73–75]. Starting from the observation of the outside of the three pyramids of the Giza plateau, for the first time, we were able to establish that the three pyramids of Khnum-Khufu, Kefren and Menkaure have eight sides. This feature, known only for the larger pyramid, is now extended to the other two. According to the authors of [73], the idea that the pyramids of the Giza plateau had this characteristic is due to the need to convey, in an orderly manner, the water that flowed along the faces of the pyramidal structures. In the case of the pyramid of Khnum-Khufu, which we have analyzed in depth, it can be assumed, in analogy with other authors [5,25,76], that it was surrounded by an enormous basin full of water, which allowed the circulation of some boats. These boats were used by some attendants with the task of bringing the water to about 90 m high, pouring it into the south shaft by using many rotating stones probably similar to the Sabu diorite stone [28]. The SAR tecnique allows us to provide evidence that the shape of this monument does not resemble a perfect pyramidic form because of the presence of a double changing in slope: the first of which is 14.5 ca. degrees at approximately 20 m high, while the second one is 6 degrees ca. at approximately 100 m high. The Nile River's water should have filled the basin up to the height of the first change of slope of the pyramid, thus allowing the Egyptian boats not to get stuck with the keel on the side of the pyramid itself. The water would have invaded the King's chamber, but having reached the height of the granite basin inside the chamber (often referred to as the sarcophagus), it would not have exceeded that level in height and would have instead risen in the north shaft, whose entrance is placed at the same height as the basin, creating an air seal that effectively airlocked the room. Having the King's chamber in fact hermetically sealed would have caused excess water to rise up the north shaft. The Queen's chamber would also be filled with water, up to the height of the shafts, by means of two connections to the shafts of the King's chamber, which were probably located in rooms 19 and 11, building a closed circuit, which is called Quincke's tube [26]. As also proposed by other authors [77], the pyramid, with its megalithic structure, was placed in vibration by the wind and the low frequencies thus developed, which acted as a low-pass filter allowing only low frequencies to bounce back on the roof of the Zed toward the King's chamber [16]. Such a room would behave like an air-filled bottle of Helmholtz [29], in which the granite basin acted as a bottleneck. The walls of the basin, vibrating at low and precise frequencies, linked to the internal and external measurements of the basin itself, proportional to multiples of $\pi$ and the Golden Ratio $\phi$ [16], would have caused the water contained in the Quincke's circuit to vibrate. These frequencies, traveling through the closed circuit of Quincke's tube, at about 1400 m/s (speed of sound in the water), reached the Queen's chamber, where the height of the water

could not exceed the height of shafts from the floor. A particular frequency could be developed, which was suitably amplified by the correct dimensions of the niche present in the west wall, which acted as a sound box for a musical instrument, releasing into the air a sound frequency that was able to interact with a cylindrical container placed on the floor of the room, the traces of which are still visible [14]. This cylindrical container, probably made of wood, was put into resonance by the obtained low frequency. Two individuals were placed both in the basin of the King's chamber and in the cylindrical container, in the Queen's chamber and appropriately treated with this low sound frequency for curative and religious purposes [21]. At the end of the procedure, the King's chamber was emptied by letting the water out of the Great Gallery and conveying it toward the room called "Grotto" toward the "Unfinished" chamber which brought the water back through a path in the floor, now occluded by debris, to the Nile. Subsequently, the Queen's chamber was emptied in two steps: first, a granite "plug" in the corridor leading to the room was removed: (this passage actually has a slight hydraulic slope toward the Great Gallery) and the water was made to flow out, at the floor of the Great Gallery, where it was conveyed toward the "Grotto". Subsequently, a plug placed in the floor was removed to finish the emptying of the room. The water thus conveyed through the hole in the floor, highlighted in a book published in 1877 [14], allowed the liquid to enter the room which, in our 3D reconstruction, corresponds to the number 14, eventually reaching the "Unfinished" room and returning to the Nile. The "Grotto" and room 14 are, in our opinion, necessary to stop the fall of water by slowing down its speed, with a mechanism similar to a common water jet pump used in laboratories to create vacuum in equipment, which is called Venturi's tube. The evident traces of erosion due to water inside the pyramid rooms are in support of our interpretative hypothesis. The three boulders that today are wedged at the beginning of the oblique corridor leading to the Great Gallery would have been used as "plugs" to block the access of water to the exit of the pyramid or from the Queen's chamber by making them flow in different positions as needed. The existence of passage 18 seems to be related to a little open room, which has never been described by anyone but is well tracked by photographic evidence, that appears located at the top of the entrance of the Great Gallery and was probably employed as security exit. The entire system of the ramps highlighted by the SAR could be interpreted as a gigantic resonant structure, having the purpose of equalizing any differences in vibration between the north and south part of the pyramid, with the aim of making the square structure reach number 10, placed around the Zed, an equalized vibrational signal. Similarly, the complex structure number 9 identified immediately below the plane on which the pyramid rests has a shape similar to structures used to absorb the effects of mechanical vibrations that are transmitted through the ground [30]. The technique proposed in this article, unlike the classical SAR tomography developed in [52,64,65], has penetration properties orders of magnitude greater because what is proposed uses the vibrations (the phonons) and not the photonic information. A disadvantage of the present technique could be the processing time needed to carry out vibrational raw data synthesis, which requires substantial computing resources. As an example, to collect a full resolution tomogram, e.g., the one in Figure 46b, using a DELL i7 PC with 32 GB of RAM installed, took approximately 6 days of computational time.

## 7. Conclusions

In this paper, we have shown how it is possible to use SAR MM Doppler tomography in an advantageous, economical, non-invasive and rapid way to make a valid contribution in the study of the structure of ancient megalithic monuments such as the pyramid of Khnum-Khufu. We are aware that only by confirmation on the field of our findings can we validate our hypothesis. However, it seemed logical to provide a hypothetical interpretation based on the data we collected that could serve as a starting point for future research.

From our point of view, the results obtained on the pyramid of Khnum-Khufu must be considered in their entirety (globally and not locally), thus taking into account the new structures found within it, the existence of which was previously unknown, all of which are

connected. We have also discovered and measured that the pyramids of the Giza Plateau present, on all faces, spatial indentations that tend to divide them in two with respect to the axes of central symmetry, constituted by their vertices. In this context, it seems clear that the multitude of structures, together with the octagonal feature of Khnum-Khufu, are to be seen as connected features. The numerous structures thus seem to belong to a gigantic resonator with the Zed that, in our personal opinion, could function as a high-order (probably fifth-order) low-pass acoustic filter due to its multi-layer and hence multi-stage characteristic. At the time of the construction of the pyramids, the Nile most probably reached the Giza plain, and the pyramids were probably flooded with water up to a couple of meters from their base. This explains why rowing boats were found without masts to support the sails. Once we have consolidated this article, we will need to provide an add-on explaining in detail the function of each chamber. At this point, it might not be at all wrong to assume that many of the structures inside the pyramid could have been purposely flooded several times a year and thus left with the liquid flowing inside. The excess water could then be expelled from the shafts, which then, thanks to the distortions on the faces (those estimated from the external results), acted as conveyors, so that the water could be transported by gravity to the ground in an orderly manner. By looking at the estimated tomographs, we were able to realize that the inner areas could be reached safely from a precise point identified by the entrance corridor on the north side (Tag 17). From our point of view, it is possible to open a minimally invasive passageway, which is the way to the exploration of all structures surveyed. This could be completed simply by drilling a passage from the main entrance (i.e., by drilling Tag 18).

In addition, the work carried out on Khnum-Khufu will certainly be extended to look inside other megalithic structures, even those belonging to other continents. At present, the research team is carrying out the same work as for Khnum-Khufu, but extended to the pyramid of Khefren, in order to have a more complete overview of the megalithic apparatus of Giza. We are confident that we will also provide Khefren's internal results in the near future. In conclusion, through these discoveries, we believe it is appropriate for the Egyptian Authorities to begin a campaign of excavations and drilling in order to bring to light and give back to all humanity all the structures we estimated through the use of the proposed method.

In the near future, we would like to extend the SAR methodology to the investigation of the internal structure of other important monuments of the Giza plateau.

**Author Contributions:** The authors contributed to all parts of this work. All authors have read and agreed to the published version of the manuscript.

**Funding:** This research received no external funding.

**Data Availability Statement:** Not applicable.

**Acknowledgments:** We would like to thank Daniele Perissin for making the SARPROZ software available, through which many calculations were carried out more easily and quickly. We also thank the Italian Space Agency for providing the SAR data. We would also like to thank Riccardo Garzelli for conducting a deep revision of the English language and for having made crucial revisions to the entire article's structure and layout. The signal processing technique presented in this work has been submitted in patent application in 4 July 2022, to Commerce Department of Malta, Industrial Property Registrations Directorate, patent application number 4451. Additional information on the technique used and consultation of other experiments can be found at www.harmonicsar.com (accessed on 1 July 2022).

**Conflicts of Interest:** The authors declare no conflict of interest.

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
