# Peer review of "Synthetic Aperture Radar Doppler Tomography Reveals Details of Undiscovered High-Resolution Internal Structure of the Great Pyramid of Giza"

_remotesensing, doi:10.3390/rs14205231_

Round 1

Reviewer 1 Report (New Reviewer)

This work describes an imaging method based on the analysis of micro-movements on the internal structure of Khnum-Khufu Pyramid using SAR Doppler Tomography.

The paper is interesting and detailed, but in my opinion, it is a bit dispersive and validation of the results is missing.

MAJOR COMMENTS

At the beginning, the purpose of your work is not well explained, what is new compared to the literature? What are you demonstrating? It is better to write it clearly in the introduction.

Page 5: “SAR interferometry, the phase differences between two different sensor positions are used to estimate the terrain topography. In this paper, The successful experimental realization of polarimetric airborne SAR tomography is demonstrated for the first time in [64].”. I think you should rewrite this sentence as it is not clear if you are still presenting the state of art or what you are proposing in your paper. Moreover, T should be in lowercase.

What I expect to find in the Methodology section is at least a scheme of what you developed and which are the steps that conducted you to the results. Please organize better this section, also adding a flowchart.

Section 5 begins by saying that you performed an InSAR procedure to analyse the fringes. This was never described in detail in the methodological section. You say at page 6: “we processed several SAR images observed in the Vertical-Vertical (VV) polarization, and the estimated MM allows us to visualize the principal internal components present in the pyramid.”. How many images? Which sensor have you used (Sentinel, COSMO-SkyMed….)? You write these information in the results section but it is better to put and expand them before the methodology. Have you pre-processed the images? How was the interferometric process performed? These information would help other scientists replicating your experimentation.

The conclusions are poor and weak, especially from the point of view of the validation.

MINOR COMMENTS

In the abstract: “The results obtained prove to be very promising” à “The obtained results prove to be very promising”.

Page 5: the second red line: “when” should be lowercase

Page 5: “This propriety depends on the used wavelengths, witch is usually pour”. Typing error “which”.

In section 1.3, state to the title, I would expect to read information about SAR, what it is, the typology of the available products and so on. Instead I mostly read how SAR has been used in literature for archaeological sites. So maybe it is better to change the section title or incorporate it into section 1.2.

In figure 2, I would add the description of each number also in the caption for a clearer comprehension.

Page 9: “Figures 4, 5, and 6 represent the time-domain displacement in magnitude, range and azimuth respectively. the blue plot is the unfiltered temporal displacement trend while the red function represents its positive envelope”. “The blue plot” should have capital letter.

Page 18: There is incoherence between the number of figures and the face of the pyramids. For example, you write “Figure 12 (a), (b), (c) and (d), are the particular representations of the SAR interferometric fringes observed on the North face of the Pyramid” but if I look at figure 12, the caption says “Interferometric phases. (a): East side.”. The same for the following images. Please correct.

In the last row of page 18, I think the figure number is missing. I suppose it is figure 18.

Section 5.2, first sentence: “Data analysis obtained using te SAR tomographic”, te àthe

In figure 21, it is better to write that table 3 lists all the numbers in the figure.

Right before section 5.4, there is “:” instead of “.”

Page 39: “thus allowing the Egyptian boats not to get stuck with the keel on the side of the pyramid itself The water” a point is missing between “itself” and “The water”.

The conclusion paragraph should stay before the Acknowledgments and, in my opinion, they should be extended.

Author Response

See attached file
Thank you

Reviewer 2 Report (New Reviewer)

this is very well done.  The equations put me off a little and I have no idea what to make of those.  But the modeling was super. Good job.  

Author Response

Thank you very much for your quotation.

Sincerely

Filippo Biondi and Corrado Malanga

Reviewer 3 Report (New Reviewer)

The proposed article is undoubtedly ambitious because it proposes an analysis of the internal structure of the Pyramid of Giza by means of satellite SAR tomography using a new method based on vibrations (phonons) instead of photon information. The authors demonstrate with appropriate calculations the goodness of the method (the signal processing system has in fact even been presented for patenting in recent months). The method is also well described from the maths point of view. Of course, the validity of the method can only be verified in reality with targeted excavations (which appear difficult to carry out) or better by using appropriate geophysical methods. In fact, in the general description, the authors do not indicate the dimensions of the pyramid and of the King and Queen's chambers, just as they do not indicate the dimensions of the blocks that cover the "ground" terrain.  These blocks are the objects responsible for the detectable vibrations and it is difficult to think that there aren't terrestrial methods capable of describing, even quantitatively, the internal parts enclosed by a surface, if not for the insufficient instrumental sensitivity which instead does not occur from satellite measurements. It must be assumed that the blocks, at times, are decoupled from the soils surrounding the internal structures and could have generated abnormal vibrations compared to those observed.

However, the article is rigorous and seems to indicate a new scientific approach that deserves to be pursued. It therefore deserves to be accepted even if with the minor revisions indicated

Suggestions

1) In the general description, the last part of paragraph 1.1, concerning the correlation with the brain waves of living creatures, seems out of context. It is proposed to eliminate it

2) It would be advisable to perform (in the future) the analysis on sections gradually farther away from the vertex with the same orientation. for a sort of 3D tomography and on other megalithic structures known in the different parts.

Author Response

Please see attached file.
Thank you

This manuscript is a resubmission of an earlier submission. The following is a list of the peer review reports and author responses from that submission.

Round 1

Reviewer 1 Report

The paper uses an Innovative approach and is well written. However there are few observations

What are factors causing and affecting the vibrations which are captured needs to be highlighted, Uncertainty  in estimation the anomalies from vibrations if any  

How the vibrations are linked to the anomalies

In this study SLC product is used, would the results be improved if one uses the RAW data as one has better control on the parameters using raw data

Reviewer 2 Report

The idea and proposed technique presented in this paper is innovative and interesting, and I'm sure that when properly investigated and presented it would be of great interest to the SAR community and to other disciplines that would benefit from non-invasive remote sensing through materials.

However, unfortunately, in my opinion the quality of presentation and of the analysis of the results is flawed and requires a significant amount of work before it is of publishable quality. I would add that the analysis presented does not in my opinion provide firm conclusions as to what is being observed in the tomograms, for example whether the tomogram features represent real objects / cavities or whether these are just artefacts of the technique.

This is a very long paper, at 40 pages, and it can be seen that the document has been put together in a rush, with many mistakes present in both figure and table captioning, and also in their cross-referencing in the text. Unfortunately this makes the lengthy analysis confusing and difficult to follow.

In addition, although I did not completely check the references list, I found errors in the list, for example a repetition of a reference (57 and 59), and it would be almost certain that there are additional mistakes in the reference list should I go through them in detail.

The proposed micro-motion tomographic method itself sounds promising, however I am left wondering what size (amplitude) of vibrations would be necessary for the technique to provide useful tomograms for a given radar frequency. Could the required amplitudes of motion be stated please? I am also unaware how the vibrations map what is vertically below the surface as opposed to what is normal to the surface tangent for example. What about effects from structures to the side of the pixel being examined. What about effects due to reflecting acoustic waves within the structures, which may be described as multipath? None of these effects are mentioned and should be given some consideration in the paper.

More importantly, for such a novel technique, I would have expected to see some careful validation experiments on well-known targets, in a well-known environment, with well characterized vibrations. The effectiveness of the approach needs to first be demonstrated before tackling the mysterious pyramids problem. Within the references, there are no other examples of this technique being applied in such a controlled manner to provide good quality tomograms. Hence there is immediately a lack of confidence of the accuracy of the technique.

The analysis includes an external to pyramids standard interferometric approach, and the new internal to pyramids micromotion-analysis approach.

Regarding the external to pyramids standard analysis, the authors should clarify the methodology: What process was employed for fitting the white lines to the interferogram? Presumably fitting by eye? What is the scale of the interferometric zoomed figures? It would have been simple to provide figure axis units of meters. What is the degree of deviation of the pyramid sides from flat planes? (e.g. 1m?). What might the errors be? None of this is covered, so cannot be regarded as a "rigorous" analysis. Perhaps the analysis may count as a basic qualitative observation with tentative results. For a good proportion of the presented results, the hypothesized facets are not actually clearly evident. Personally, I could agree from the evidence that there may be some kind of "sagging" process occurring on the pyramid structures, but with the evidence and analysis presented I would not be able to support the conclusion that all the pyramids have eight facets. I would therefore suggest either a more rigorous presentation of method and results, or a moderation of the conclusion statements here. A standard approach to such a problem, often used in archaeology, is airborne LIDAR, which would provide accurate results for comparison.

Regarding the internal to pyramid analysis, none of the tomograms have meaningful axes, providing an indication of size, for example in metres. The tomograms are overlayed with a pyramid sketch, leading to the impression that the authors have stretched and translated the tomograms and the sketch until by eye, they give the impression of a possible match in features. Even after the authors have declared a match with known features, it is often difficult to agree with their conclusions as to what is being represented, and whether there is a close match with the known features. The analysis of the known features are important, as these constitute the method control experiment, and without this control experiment analysis being well conducted, the rest of the conclusions are brought into doubt.

As a minimum, I would insist that the tomograms have meaningful axes and labels, say in meters, and that when an overlay is presented, the overlay should be as much as possible automatic (without the need for arbitrary user stretches and translations), with a transparent suitable projections of their proposed CAD model, rather than with the standard sketches. This would be especially useful for the “new” yet unknown pyramid feature hypotheses. This type of analysis can be easily carried out in common scientific programming environments such as Matlab. Where possible, the alignment of CAD rendering and tomogram should be automated, to not be reliant on subjective ad hoc alignment and scaling from the authors, which could lead to erroneous and non-supported conclusions.

Have the authors considered obtaining 3D renderings of volumetric information from within the pyramid, these would be more convincing.

In addition to the comments provided above, as part of the review, please include the annotated pdf file uploaded here, which includes additional review comments.

Author Response

Please consider the attached file.

Reviewer 3 Report

In this study, authors present an internal structure mapping method using the synthetic aperture radar Doppler Tomography. The 3D tomographic imaging of the Khnum-Khufu Pyramid was achieved. Although the concept of this work is interesting and innovative, the manuscript needs to be major revised to facilitate a better understanding and readability. The detailed comments are in following:

1) The introduction needs to be more concentrated on the scope of this study. In this version, it is difficult to grasp the logic of this study, such as from the raised problem, to the state-of-the-art the technology and application, and then the proposed solution.

2) Methodology section: what is the advantage of the proposed 3D tomography method in this study compared to the one applied in [54]. More clarification on this issue is needed.

3) The captions of Figures are wrongly given or misleading. There are no clarifications on sub-captions of (a), (b) and (c), etc.

4) It is contradictory for the data description applied in this study. For instance, it is said the Stripmap of COSMO SkyMed SAR imagery was utilized (Table 1), however, in the page 7, the authors described that the staring-spotlight SAR acquisition was adopted.

5) What is Gsa ? No corresponding notation can be found in Figure 3.

6) Figures 6-15 from repeated cycle InSAR results are misleading and out of the scope of this study. Thus, they need to be removed.

7) External Experiment Results can be removed in order to pinpoint the 3D tomography as well as the result interpretations.

8) The advantage and limitation of 3D SAR tomography needs to be further discussed in the Discussion section.

9) Acknowledgments and Author contribution statements need to be added after the Conclusion section.

In summary, the manuscript in this version needs to be carefully edited and reorganized for a better readability as well as to further pinpoint the novelty either from the technology or archaeology application perspective.

Author Response

Please Consider the attached file.
